# Let's Be Self-generated via Step by Step: A Curriculum Learning Approach to Automated Reasoning with Large Language Models

## Abstract

While Chain of Thought (CoT) prompting approaches have significantly consolidated the reasoning capabilities of large language models (LLMs), they still face limitations that require extensive human effort or have performance needs to be improved. Existing endeavors have focused on bridging these gaps; however, these approaches either hinge on external data and cannot completely eliminate manual effort, or they fall short in effectively directing LLMs to generate high-quality exemplary prompts. To address the said pitfalls, we propose a novel prompt approach for automatic reasoning named **LBS3**, inspired by curriculum learning which better reflects human learning habits. Specifically, LBS3 initially steers LLMs to recall easy-to-hard proxy queries that are pertinent to the target query. Following this, it invokes a progressive strategy that utilizes exemplary prompts stemmed from easy-proxy queries to direct LLMs in solving hard-proxy queries, enabling the high-quality of the proxy solutions. Finally, our extensive experiments in various reasoning-intensive tasks with varying open- and closed-source LLMs show that LBS3 achieves strongly competitive performance compared to the SOTA baselines. Our code is here: https://anonymous.4open.science/r/LBS3-B926.

## 1 Introduction

With super-sized training corpora and computational cluster resources, Large Language Models (LLMs) have demonstrated *emergent capabilities*, thus enabling state-of-the-art performance in a wide range of natural language tasks (Wei et al., 2022a; Brown et al., 2020; Chowdhery et al., 2023; Liang et al., 2022; Qin et al., 2023; Wei et al., 2023; Touvron et al., 2023). However, directly applying LLMs to complex reasoning tasks (e.g., mathematical reasoning, commonsense reasoning, etc.) in a naive manner presents significant challenges (Yin et al., 2023b; Wei et al., 2022b; Kojima et al., 2022). For instance, the performance may be inadequate when simply feeding queries or using few-shot query-answer pairs in in-context learning (ICL) approaches for these kinds of tasks. Recent studies have shed light on that prompting LLMs to generate multiple reasoning steps (i.e., rationale) can markedly enhance their ability to reason, resulting in the development of the chain-of-thought (CoT) prompting (Wei et al., 2022b; Kojima et al., 2022; Zhou et al., 2022b; Wang et al., 2022b; Aggarwal et al., 2023a; Chen et al., 2024a; Yao et al., 2024; Zou et al., 2023; Yu et al., 2024; Besta et al., 2024). Current CoT prompting approaches fall broadly into two categories, namely Few-Shot CoT (FS-CoT) (Wei et al., 2022b) and Zero-Shot CoT(ZS-CoT) (Kojima et al., 2022). Among them, FS-CoT involves providing LLMs with few task-specific context exemplars of query-rationale-answer triplets tied to the target query to prompt the generation of reasoning steps; ZS-CoT instead stimulates LLMs' reasoning capabilities by furnishing general trigger instructions (such as "Let's think step by step") appended to the target query.

Despite their considerable success, obstacles persist in the field of prompt engineering research that plague real-world applications. FS-CoT, for example, delivers well-crafted exemplary prompts but at the cost of labor-intensive manual annotations. To mitigate this, some efforts have been made to enhance the quality of exemplary prompts by retrieving the most relevant, complex and diverse existing queries or exemplars for the target task, which is achieved by tapping into external sources related to the task at hand, such as datasets or corpora, and employing various pre-defined similarity metrics (Liu et al., 2021; Rubin et al., 2021; Fu et al., 2022; Ye et al., 2022; Su et al., 2022; Wu

et al., 2022; Ye & Durrett, 2023; Diao et al., 2023; Wan et al., 2023). Nevertheless, the required external sources these approaches rely on may not always be available in practice, and they may not completely obviate the need for manual labeling. Moreover, while ZS-CoT offers versatility, its performance often lags behind FS-CoT in a variety of complex reasoning tasks.

To overcome the aforementioned issues, recent initiatives (e.g., Self-ICL (Chen et al., 2023), Auto-ICL (Yang et al., 2023) and Analogical Prompting (Yasunaga et al., 2023)) work on bootstrapping LLMs to self-generate few-shot new proxy queries that are relevant and diverse to the target query, along with constructing corresponding exemplary prompts of triplets, thereby augmenting their capabilities to tackle reasoning tasks. Essentially, these methods draw parallels with the concept of analogical reasoning in psychology, where individuals rely on past related experiences to approach new problems (Vosniadou & Ortony, 1989). The underlying insight behind them is that pre-trained LLMs (such as Llama3, GPT-3.5 and GPT-4.0, etc.) have amassed a wealth of knowledge that equips them to fulfill various reasoning tasks. However, we observed that merely prompting LLMs to recall experiences related to the target queries may lead to the generation of proxy queries that are as difficult as the target queries themselves, along with corresponding incorrect proxy solutions, misguiding the resolution of the target queries, as exemplified in Fig. 1. See the related works in Appendix A for more details.

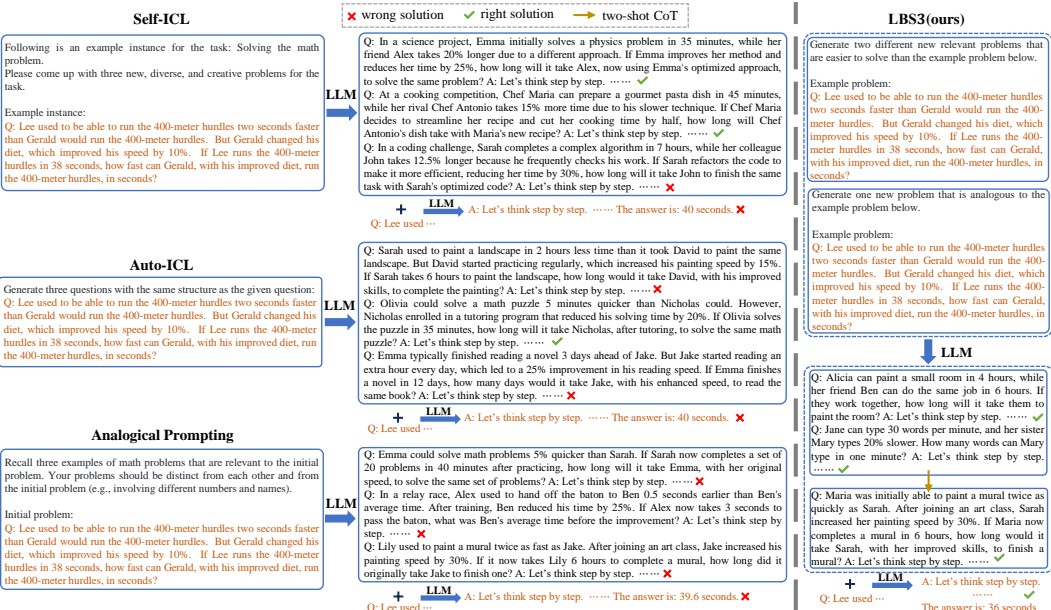

Figure 1: The illustrative comparison between LBS3 and existing representative approaches (including Self-ICL, Auto-ICL and Analogical Prompting) regarding proxy queries generated on top of Qwen1.5-14B-Chat. Given a mathematical query, i.e., "Q: Lee used ⋯ ", Self-ICL, Auto-ICL and Analogical Prompting purely explore new, diverse and relevant proxy queries. In contrast, LBS3 investigates that from easy to hard. Note that the implementation of Analogical Prompting follows the original paper, and we break down the results for ease of illustration.

The issues mentioned above motivate us to propose a novel automatic reasoning prompt approach, coined **LBS3**, which is inspired by curriculum learning that mirrors the progressive nature of human learning styles. The idea of curriculum learning (Bengio et al., 2009; Cornacchia & Mossel, 2023) has been widely applied in the field of artificial intelligence, emulating the human learning process from easy to hard tasks (Campos, 2021; Maharana & Bansal, 2022; Huang et al., 2020; Kong et al., 2021; Zhu et al., 2022; Li et al., 2021; Soviany et al., 2022; Xu et al., 2020). Thus, **LBS3 critically aims to 1) guide LLMs to generate easy- (or hard-) proxy queries related to the target query and 2) enhance the effectiveness of the solutions for these proxy queries to benefit that for the target query.** For the **former**, diverging from existing approaches (e.g., Self-ICL and Auto-ICL) that generate proxy queries in one pass, we develop a two-stage framework of generation for proxy queries. Specifically, we first prompt LLMs with instructions like "Generate $n_1$ different new relevant problems that are easier

to solve than the example problem below." to generate simpler proxy queries than the given query, that is, easy-proxy queries. Then, we instruct LLMs to formulate analogical proxy queries for the given query, which are the hard-proxy queries, by using instructions like "Generate $n_2$ different new problems that are analogous to the example problem below." Note that $n_1$ and $n_2$ denote the number of proxy queries generated. For the **latter**, we initially leverage LLMs to solve each easy-proxy query independently in the ZS-CoT manner, creating corresponding triplet exemplary prompt. Subsequently, we combine these prompts with each hard-proxy query and generate solutions in the FS-CoT fashion. Ultimately, we amalgamate all constructed exemplary prompts with the given query and derive the target solution in the FS-CoT manner. We modularly outline the generic framework of the reasoning process for LBS3 in Fig. 2.

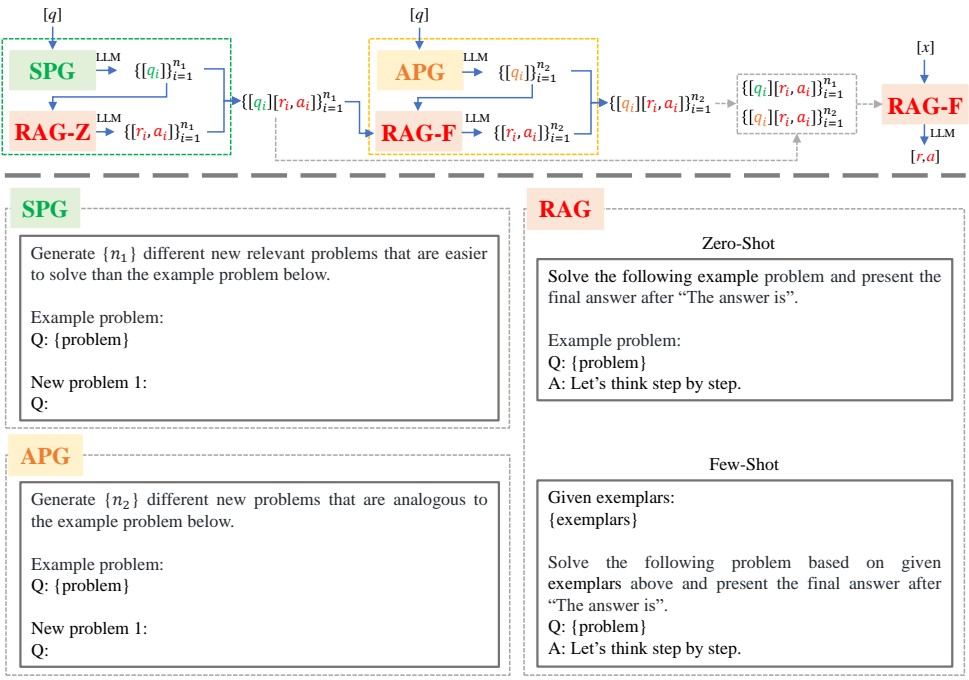

Figure 2: The overview of LBS3 approach.

One of the advantages for the proposed approach is that LBS3 explicitly distinguishes between easy- and hard-proxy queries, and ensures that the difficulty of solving proxy queries does not exceed that of the given query. Additionally, in contrast to existing approaches that tackle each proxy query from scratch, we adopt a progressive strategy to harness exemplary prompts derived from easy-proxy queries to guide the generation of solution for hard ones, thereby alleviating the accumulation of errors arises from reasoning ab initio (Yu et al., 2024). To the best of our knowledge, our work is the pioneering attempt to emulate the idea of curriculum learning, aiming to investigate how LLMs can self-generate few-shot exemplary prompts to facilitate the reasoning process.

Our main contributions of this work are summarized as follows. First, we put forward a new automatic reasoning prompt approach (LBS3), which is inspired by the idea of curriculum learning to assist LLMs in recalling easy and hard proxy queries related to the target query. Second, we adopt a progressive strategy that utilizes exemplary prompts derived from easy-proxy queries to direct LLMs in solving hard-proxy queries, improving the quality of the proxy solutions. At last, we conducted extensive experiments focused on reasoning-intensive tasks. These tasks included mathematical problem-solving (GSM8K (Cobbe et al., 2021), MATH (Hendrycks et al., 2021), and SVAMP (Patel et al., 2021)), commonsense reasoning (StrategyQA (Geva et al., 2021) and CommonsenseQA (Talmor et al., 2018)), as well as reasoning tasks within BBH (Srivastava et al., 2022). Moreover, LLMs used for these reasoning tasks encompass open-source models (Qwen1.5-14B (Bai et al., 2023), Qwen1.5-72B (Bai et al., 2023), and Llama3-70B (AI@Meta, 2024)) and closed-source models[1]

---

[1]https://openai.com/api/

(GPT-3.5-turbo and GPT-4.0-turbo). Empirical results show that LBS3 is highly competitive in reasoning performance compared with state-of-the-art baselines. This underscores the effectiveness of generating tailored exemplary prompts ranging from easy to hard for a given query, significantly bolstering the reasoning capabilities of LLMs.

## 2 PRELIMINARIES

In this paper, we work on scenarios wherein we address a given query (e.g., a math problem, multiple-choice question, or true/false assessment, etc.) without any additional information. Given a query $[q]$, the objective is to produce a solution consisting of the rationale (i.e., multiple reasoning steps) and the final answer, denoted by $[r, a]$. A prompt template, represented by $T$, is designed for solving $[q]$. Note that multiple sub-prompt templates are assembled to form pipeline templates in certain specific prompt approaches. A prevalent prompting approach aims to integrate $T$ with $[q]$, resulting in $h = inte(T, [q])$, which is then fed to an LLM to elicit the corresponding solution $[r, a] = LLM(h)$. Listed below are the existing prompting approaches that are most pertinent to our work.

- In ZS-CoT (Kojima et al., 2022), $T$ and $[q]$ are integrated as $h =$"$[q]$ `Let's think step by step.`"
- In FS-CoT (Wei et al., 2022b), $n$-shot manually crafted exemplary prompts are used to form $T$, which, when combined with $[q]$, results in $h =$"$[q_1][r_1, a_1] \cdots [q_n][r_n, a_n][q]$".
- In Analogical Prompting (Yasunaga et al., 2023), the integration of $T$ with $[q]$ yields $h$, which prompts an LLM to self-generate $n$-shot distinct proxy exemplars relevant to $[q]$ and proceed to solve $[q]$, i.e., $[q_1][r_1, a_1] \cdots [q_n][r_n, a_n][q][r, a] = LLM(h)$. Of note, the one-pass generation mode employed in this approach necessitates that the LLM possesses robust capabilities for both following instructions and generating responses.
- In Self-ICL (Chen et al., 2023)/Auto-ICL (Yang et al., 2023), a two-step policy is used to steer an LLM to self-generate $n$-shot exemplars for solving $[q]$. Initially, the construction of $h_1$, as illustrated in detail in Fig. 1, directs the LLM to generate $n$-shot proxy queries, i.e., $[q_1] \cdots [q_n] = LLM(h_1)$. Subsequently, the LLM is deployed to solve each proxy query one by one, i.e., $[r_i, a_i] = LLM(h_2^i)$ ($i \in [n]$) where $h_2^i =$"$[q_i]$ `Let's think step by step.`" The process culminates with the assembly of $h =$"$[q_1][r_1, a_1] \cdots [q_n][r_n, a_n][q]$". Notably, compared to Analogical Prompting, Self-ICL/Auto-ICL, despite incurring additional computation and costs due to multiple interactions with the LLM, offer greater flexibility and more closely mirror human cognitive processes. Also, they place more modest demands on the LLM's ability to follow instructions and generate responses.

We aim to tailor a prompt approach that enables an LLM to self-generate high-quality proxy exemplars, improving the accuracy of the solution it produces for a given query $[q]$.

## 3 APPROACH

In this section, we elaborate on our approach, LBS3, which draws inspiration from the concept of curriculum learning. LBS3 empowers an LLM to self-generate few-shot exemplars that are pertinent to the target query (a.k.a. problem), ranging from simple to complex. Figure 2 illustrates the reasoning pipeline of LBS3 in a modular fashion, which contains three key modules: the Simple Problem Generation (**SPG**) module, the Analogous Problem Generation (**APG**) module and the Rationale and Answer Generation (**RAG**) module. Remarkably, the RAG consists of two sub-modules: one that solves the given query using the ZS-CoT manner and the other utilizing the FS-CoT manner, denoted as RAG-Z and RAG-F, respectively. Thereafter, we introduce LBS3 from two perspectives: firstly, how it bootstraps an LLM to generate proxy queries related to the given query in increasing order of difficulty, and secondly, how it effectively addresses the more challenging proxy queries.

### 3.1 TWO-STAGE GENERATION OF PROXY QUERIES

To enable the generation of proxy queries with varying levels of difficulty, we propose a two-stage framework. Specifically, suppose we need to generate $n$ proxy queries, comprising $n_1$ easy-proxy

queries and $n_2$ hard-proxy queries, i.e., $n = n_1 + n_2$. Also, to clearly understand LBS3 approach, we present its pseudocode as shown in Alg. 1.

In the first stage, LBS3 inputs **SPG**.format($n_1$, problem=$[q]$) into the LLM to produce the easy-proxy queries $\{[q_i]\}_{i=1}^{n_1}$ (lines 3-4); then it utilizes **APG**.format($n_2$, problem=$[q]$) to induce the LLM to generate the hard-proxy queries $\{[q_i]\}_{i=1}^{n_2}$ (lines 12-13) in the second stage. Accounting for the said process, we can explicitly and precisely control the proportion of easy- and hard-proxy queries using succinct and effective instructions, by selecting different combinations of $n_1$ and $n_2$. For instance, when $n_1 = 0$, LBS3 focuses on generating analogical (i.e., hard) proxy queries; whereas when $n_1 = n$, it only generates easy-proxy queries. Thus, it ensures that the difficulty of solving hard-proxy queries (i.e., analogical proxy queries) does not significantly exceed that of the given query $[q]$.

One might inquire whether it is feasible to design a prompt template that allows an LLM to generate $n$ proxy queries ranging from easy to hard in one go? Indeed, it is feasible. In our experiments, we use the instruction "Generate $n_1$ different new relevant problems that are easier to solve than the example problem below. And then generate $n_2$ different new problems that are analogous to the example problem below." to generate proxy queries that meet the two-stage requirements in one go. Consequently, lines 3-4 and 12-13 in Alg. 1 can be condensed into a single-step process, circumventing additional computational costs. Due to space constraints, we provide empirical examples in Appendix C.

## 3.2 PROGRESSIVE STRATEGY OF SOLVING QUERIES

Now, we propose a progressive strategy to effectively solve the aforementioned proxy queries (especially $\{[q_i]\}_{i=1}^{n_2}$). To commence, we sequentially solve each easy-proxy query in $\{[q_i]\}_{i=1}^{n_1}$ with the ZS-CoT manner, which yields $\{[q_i][r_i, a_i]\}_{i=1}^{n_1}$ (lines 6-10). Then, $\{[q_i][r_i, a_i]\}_{i=1}^{n_1}$ are used as exemplary prompts to integrate each hard-proxy query from $\{[q_i]\}_{i=1}^{n_2}$ within **RAG-F** and solve them one by one in FS-CoT manner, leading to $\{[q_i][r_i, a_i]\}_{i=1}^{n_2}$ (lines 14-18). Finally, we take advantage of all the proxy exemplary prompts to solve $[q]$ (lines 20-21), which in turn leads to the final solution.

---

**Algorithm 1** The pseudocode of LBS3 approach given one query $[q]$.

---

**Input:** $[q]$: the target query, $LLM$: large language model, $n_1$: the number of easy-proxy queries, $n_2$: the number of hard-proxy queries

1: Initial modules: **SPG**, **APG** and **RAG**
2: # *Stage 1:*
3: prompt_spg=**SPG**.format($n_1$, problem=$[q]$)
4: $\{[q_i]\}_{i=1}^{n_1} = LLM(\text{prompt\_spg})$
5: exem_sa = { }
6: **for** $[q_i]$ in $\{[q_i]\}_{i=1}^{n_1}$ **do**
7:     prompt_rag-z = **RAG-Z**.format(problem=$[q_i]$)
8:     $[r_i, a_i]= LLM(\text{prompt\_rag-z})$
9:     exem_sa = exem_sa $\bigcup \{[q_i][r_i, a_i]\}$
10: **end for**
11: # *Stage 2:*
12: prompt_apg = **APG**.format($n_2$, problem=$[q]$)
13: $\{[q_i]\}_{i=1}^{n_2} = LLM(\text{prompt\_apg})$
14: **for** $[q_i]$ in $\{[q_i]\}_{i=1}^{n_2}$ **do**
15:     prompt_rag-f = **RAG-F**.format(exemplars=exem_sa, problem=$[q_i]$)
16:     $[r_i, a_i]= LLM(\text{prompt\_rag-f})$
17:     exem_sa = exem_sa $\bigcup \{[q_i][r_i, a_i]\}$
18: **end for**
19: # *Stage 3:*
20: prompt_rag-f = **RAG-F**.format(exemplars=exem_sa, problem=$[q]$)
21: $[r, a] = LLM(\text{prompt\_rag-f})$

**Output:** $[r, a]$

---

The primary advantage of the above strategy is its effectiveness in enhancing the solutions for hard-proxy queries. To be specific, the easy-to-solve $\{[q_i]\}_{i=1}^{n_1}$ ensures that the corresponding exemplary prompts may be correct with high confidence. Meanwhile, (Chen et al., 2023; Yang et al., 2023; Yasunaga et al., 2023) have shown that using solved proxy queries related or analogous to $[q]$ as exemplary prompts can effectively improve the solution for $[q]$. However, when the difficulty of solving query $[q]$ is high, the generated proxy queries are likely to have comparable challenging, resulting in low-quality exemplary prompts. Therefore, adopting our proposed progressive strategy can alleviate the cumulative errors associated with solving hard-proxy queries from scratch (Yu et al., 2024), thereby enhancing the quality of their solutions. **In addition, we find that using already solved hard-proxy queries as additional exemplary prompts for solving the next hard-proxy query can further strengthen the solution to** $[q]$, see Alg. 1 and Section 4.3.4 for more details and empirical justification.

## 4 EXPERIMENTS

### 4.1 EXPERIMENTAL SETTINGS

**Datasets and LLMs.** In this section, we empirically investigate LBS3 on eight benchmarks commonly utilized for reasoning tasks, spanning three categories of reasoning tasks: (i) mathematical problem-solving (GSM8K (Cobbe et al., 2021), MATH (Hendrycks et al., 2021), and SVAMP (Patel et al., 2021)); (ii) commonsense reasoning, as seen in StrategyQA (SQA) (Geva et al., 2021) and CommonsenseQA (CQA) (Talmor et al., 2018); and (iii) selected reasoning tasks within BBH (Srivastava et al., 2022), including logical deduction five objects (BBH-idfo), reasoning about colored objects (BBH-raco) and temporal sequences (BBH-ts). It is worth noting that the selected dataset mentioned above draws upon the datasets used in existing works (Yasunaga et al., 2023; Chen et al., 2023; Yang et al., 2023). Also, we resort to the five latest and most robust LLMs to perform the aforementioned reasoning tasks, which includes both open source models—Qwen1.5-14B-Chat (Bai et al., 2023), Qwen1.5-72B-Chat (Bai et al., 2023), and Meta-Llama-3-70B-Instruct (marked as Llama-3-70B-Instruct) (AI@Meta, 2024) (see Appendix B for computing devices and platforms)—as well as closed-source models accessed through the OpenAI API[2], namely gpt-3.5-turbo-instruct (marked as GPT-3.5-turbo) and gpt-4-turbo-2024-04-09 (marked as GPT-4.0-turbo) (Ouyang et al., 2022; Achiam et al., 2023).

**Baselines and Configurations.** We compare the five most relevant existing approaches to our work as baselines: Few-shot CoT(FS-CoT) (Wei et al., 2022b), Zero-shot CoT(ZS-CoT) (Kojima et al., 2022), Analogical Prompting (Ana-Pro) (Yasunaga et al., 2023), Self-ICL (Chen et al., 2023), and Auto-ICL (Yang et al., 2023). Please refer to Section 2 for more details. To ensure fairness, we employ an equal number of CoT exemplars for all approaches across models and benchmarks, regardless of whether they are manually crafted exemplars or generated proxy exemplars. Specifically, we set the number of exemplars $n$ to 4 for MATH and SVAMP benchmarks, while for the remaining benchmarks, we establish $n = 3$. In our proposed approach, LBS3, we default to setting $n_1 = 2$ and $n_2 = n - n_1$ unless stated otherwise. Furthermore, during the decoding process, we employ a greedy search algorithm for open source LLMs to generate solutions. And for closed-source models, due to randomness, we report the average of the results from three runs.

### 4.2 RESULTS COMPARISON

We explore the performance of different approaches on varying benchmarks and LLMs in terms of accuracy, with the complete results reported in Table 1. From Table 1, it is evident that LBS3 consistently outperforms the baselines with respect to average accuracy for all LLMs. Specifically, compared to the second-best baselines, LBS3's average accuracy improved by 2.49% on Qwen1.5-14B-Chat, 2.83% on Qwen1.5-72B-Chat, 2.89% on Llama3-70B-Instruct, 3.44% on GPT-3.5-turbo, and 4.30% on GPT-4.0-turbo. Intuitively, the effectiveness of LBS3 in various reasoning benchmarks becomes more pronounced with the more capable LLMs. The results demonstrate that LBS3 is suitable for varying LLMs and exhibits robustness and versatility in handling various reasoning tasks. We attribute the performance advantage of LBS3 to its effective two-stage framework for

---

[2]https://openai.com/api/

self-generating proxy queries from easy to hard, and to the progressive strategy employed to solve them. Thereafter, we delve deeper into the efficacy of these two key components in ablation study.

Furthermore, the baselines Self-ICL and Auto-ICL uniformly beat ZS-CoT in terms of average accuracy and surpassed FS-CoT in most cases. This result highlights that guiding LLMs to autonomously generate proxy exemplars relevant to a given query can effectively improve their reasoning capabilities. Additionally, the baseline Ana-Pro consistently underperforms other competitors w.r.t. average accuracy, including ZS-CoT, on open-source LLMs, yet consistently outstrips ZS-CoT w.r.t. average accuracy on closed-source LLMs. The said result confirms the high requirements imposed by the Ana-Pro approach on LLMs for following instructions and generating responses. It is worth noting that the open-source model Llama3-70B-Instruct achieves the best accuracy across all prompting approaches and even shows significant performance gains compared with the closed-source model GPT-4.0-turbo in most cases. We speculate that this phenomenon occurs because we only considered a limited set of reasoning tasks, and thus closed-source models may show stronger generalization capabilities on more and wider tasks.

Table 1: Performance comparison of different approaches in terms of accuracy (%) on various benchmarks and Large Language Models (LLMs). Note that Avg. denotes the average accuracy across various benchmarks using distinct baselines and LBS3.

|  | GSM8K | MATH | SVAMP | SQA | CQA | BBH-ldfo | BBH-raco | BBH-ts | Avg. |
|---|---|---|---|---|---|---|---|---|---|
| **Qwen1.5-14B-Chat** | | | | | | | | | |
| FS-CoT | 78.7 | 36.8 | 84.4 | 62.8 | 69.6 | 54.8 | 71.8 | 53.2 | 64.01 |
| ZS-CoT | 77.9 | 28.9 | 80.0 | 59.8 | 67.2 | 49.2 | 67.2 | 50.8 | 60.12 |
| Ana-Pro | 75.1 | 29.9 | 80.2 | 61.8 | 66.6 | 42.0 | 63.6 | 50.4 | 58.70 |
| Self-ICL | 80.7 | 38.1 | 82.2 | 64.8 | 68.8 | 56.8 | 74.8 | 51.2 | 64.67 |
| Auto-ICL | 79.3 | 37.4 | 81.8 | 63.4 | 67.8 | 50.4 | 73.6 | 52.8 | 63.31 |
| LBS3 | **81.3** | **40.8** | **85.8** | **67.8** | **70.4** | **58.4** | **75.6** | **57.2** | **67.16** |
| **Qwen1.5-72B-Chat** | | | | | | | | | |
| FS-CoT | 87.4 | 46.0 | 88.6 | 73.6 | 81.6 | 62.0 | 81.2 | 52.2 | 71.58 |
| ZS-CoT | 83.0 | 43.3 | 87.0 | 70.6 | 77.2 | 54.8 | 78.8 | 51.6 | 68.29 |
| Ana-Pro | 84.6 | 45.0 | 87.0 | 75.0 | 78.0 | 59.6 | 50.8 | 43.6 | 65.45 |
| Self-ICL | 88.0 | 50.0 | 88.2 | 78.0 | 80.4 | 60.8 | 83.2 | 53.6 | 72.78 |
| Auto-ICL | 88.6 | 48.1 | 88.0 | 76.6 | 81.4 | 64.4 | 86.0 | 53.2 | 73.29 |
| LBS3 | **88.8** | **53.1** | **91.0** | **83.2** | **82.4** | **65.2** | **86.4** | **58.8** | **76.12** |
| **Llama3-70B-Instruct** | | | | | | | | | |
| FS-CoT | 94.0 | 53.6 | 92.6 | 78.8 | 80.8 | 77.6 | 92.8 | 95.0 | 83.15 |
| ZS-CoT | 93.4 | 51.1 | 91.4 | 75.6 | 76.4 | 66.8 | 85.0 | 91.2 | 78.86 |
| Ana-Pro | 91.2 | 47.7 | 91.8 | 73.4 | 82.6 | 62.4 | 69.6 | 92.0 | 76.34 |
| Self-ICL | 93.6 | 56.6 | 91.8 | 76.6 | 79.4 | 65.6 | 90.4 | 96.8 | 81.35 |
| Auto-ICL | 94.2 | 52.9 | 90.4 | 77.2 | 79.0 | 74.4 | 90.6 | 99.6 | 82.29 |
| LBS3 | **94.6** | **59.6** | **93.6** | **80.4** | **83.6** | **78.0** | 91.6 | **100.0** | **85.18** |
| **GPT-3.5-turbo** | | | | | | | | | |
| FS-CoT | 82.1 | 45.3 | 84.9 | 74.7 | 79.3 | 45.7 | 70.5 | 79.7 | 70.27 |
| ZS-CoT | 81.3 | 44.3 | 81.9 | 69.9 | 72.5 | 39.9 | 66.7 | 73.9 | 66.29 |
| Ana-Pro | 82.1 | 48.0 | 84.3 | 72.1 | 78.3 | 47.5 | 68.7 | 74.3 | 69.41 |
| Self-ICL | 85.3 | 47.1 | 83.9 | 77.1 | 80.3 | 46.5 | 71.3 | 77.7 | 71.15 |
| Auto-ICL | 81.6 | 48.7 | 82.4 | 75.2 | 80.8 | 46.8 | 69.6 | 79.6 | 70.59 |
| LBS3 | **87.6** | **50.1** | **87.0** | **78.4** | **83.0** | **54.6** | **73.4** | **82.6** | **74.59** |
| **GPT-4.0-turbo** | | | | | | | | | |
| FS-CoT | 92.8 | 48.9 | 85.6 | 85.2 | 81.2 | 69.2 | 77.2 | 87.2 | 78.12 |
| ZS-CoT | 90.3 | 48.4 | 83.0 | 78.8 | 76.0 | 57.6 | 74.8 | 86.0 | 74.36 |
| Ana-Pro | 93.4 | 52.3 | 84.5 | 79.2 | 84.0 | 63.6 | 76.0 | 90.0 | 77.87 |
| Self-ICL | 94.5 | 54.2 | 88.2 | 80.4 | 82.8 | 68.8 | 77.2 | 91.6 | 79.71 |
| Auto-ICL | 93.6 | 53.6 | 86.9 | 82.0 | 84.6 | 71.2 | 75.6 | 93.2 | 80.08 |
| LBS3 | **94.9** | **64.2** | **89.8** | **86.6** | **86.0** | **74.8** | **82.8** | **96.0** | **84.38** |

## 4.3 ABLATION STUDY

In this section, we carefully demonstrate the efficacy and indispensability of the core components in our approach on Qwen1.5-14B-Chat, Llama3-70B-Instruct, and GPT-3.5-turbo over diverse benchmarks. These components include the two-stage framework for self-generating proxy queries from easy to hard, as well as the progressive strategy employed to solve them.

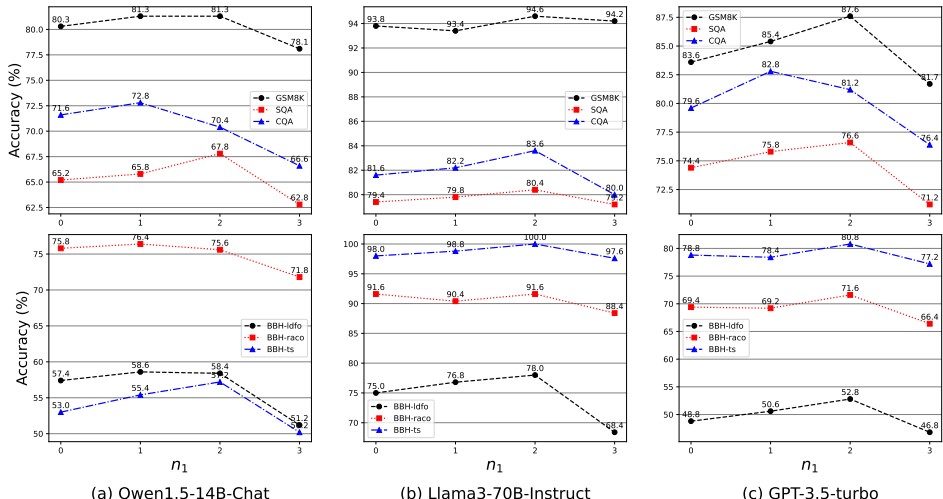

Figure 3: Accuracy (%) of LBS3 with varying $n_1$ over GSM8K, SQA, CQA, BBH-ldfo, BBH-raco and BBH-ts benchmarks.

### 4.3.1 COMPARISON OF THE NUMBER FOR EASY AND HARD PROXY EXEMPLARS

We look into the impacts of different hyperparameter combinations $(n_1, n_2)$ within the two-stage framework for self-generating proxy queries of LBS3 across various benchmarks, including GSM8K, SQA, CQA, BBH-idfo, BBH-raco, and BBH-ts. For clarity, assume that the number of proxy exemplars $n$ is 3, with both $n_1$ and $n_2$ taking values from $\{0, 1, 2, 3\}$. Note that since $n = n_1 + n_2$, we opt to only consider $n_1$, then $n_2 = n - n_1$. When $n_1 = 1$ and $n_2 = 2$, this indicates that LBS3 is tasked with generating one easy-proxy exemplar and two hard-proxy ones. We report the performance of LBS3 corresponding to varying $n_1$ on different models and benchmarks in Fig. 3.

It can be observed from Fig.3 that the accuracy of LBS3 improves with the increase of $n_1 \in \{0, 1, 2\}$ in most cases, and that the performance of LBS3 at $n_1 = 3$ consistently underperforms its performance at other $n_1$ values across all cases. To be specific, the top accuracy is achieved for LBS3 in 13 out of 18 cases when $n_1 = 2$ and is improved by an average of 1.25% compared to $n_1 = 1$ which achieves sub-optimal accuracy. This suggests that compared to the case where only easy- or hard-proxy exemplars are generated, LBS3 has superior performance when they are both present. To put it differently, LBS3, drawing on the principle of curriculum learning, effectively enhances the abilities of LLMs to handle complex reasoning tasks. Also, we see from Fig. 3 and Table 1 that LBS3's accuracy at $n_1 = 0$ (i.e., generating only hard-proxy exemplars) consistently exceeds that at $n_1 = 3$ (i.e., generating only easy-proxy exemplars) with an average improvement of 2.75%. Additionally, LBS3's accuracy at $n_1 = 3$ consistently surpasses that of ZS-CoT by an average of 2.91%. This suggests that easy-proxy exemplars provide a weaker performance boost to LLMs than hard-proxy exemplars. We posit that this is because, although hard-proxy exemplars may come with lower-quality solutions, they help LLMs to recall more useful information, whereas the opposite is true for easy-proxy exemplars. Therefore, the primary utility of the easy-proxy exemplars is to augment the quality of solutions in the hard-proxy exemplars.

### 4.3.2 STUDY FOR EASY- AND HARD-PROXY QUERIES

Based on Section 4.3.1, we further investigate the quality of proxy queries generated by SPG and APG prompt modules and their solution accuracy. To this end, we select the generations of Llama3-70B-Instruct over benchmarks GSM8K, SQA, and BBH-raco. However, determining the quality of generated proxy queries and the correctness of their answers is challenging. On the one hand, there is a lack of effective criteria for dividing the difficulty of generating proxy queries, and on the other hand, there are no plug-and-play standard answers to evaluate the solutions of proxy queries. To overcome the above difficulties, we combine GPT-4o[3] and human evaluation to study 50 randomly sampled

---
[3]https://openai.com/api/

target queries in each experiment. Specifically, we first use GPT-4o as a discriminator to score the difficulty of proxy queries generated by SPG and APG in LBS3, with target queries as a reference, following the scoring rules detailed in Table 3 of Appendix D. Then, we invite 10 human participants (all of whom are Ph.D. or Master students) to assess the correctness of the solutions for the generated proxy queries. After completing the above process, we report the average difficulty scores and corresponding solution accuracy of the proxy queries generated by SPG and APG respectively, as shown in Table 2.

Table 2: Quality study of easy- and hard-proxy queries. Note that, SPG/APG-diff (acc) represents the average difficulty score (solution accuracy(%)) of proxy queries generated by the SPG/APG.

| Benchmarks | $n_1$ value | num. $n_1/n_2$ | SPG-diff | APG-diff | SPG-acc | APG-acc |
|---|---|---|---|---|---|---|
| GSM8K | $n_1 = 0$ | 0/150 | - | 0.14 | - | 92.6 |
| | $n_1 = 1$ | 50/100 | -1.12 | 0.29 | **100.0** | 94.0 |
| | $n_1 = 2$ | 100/50 | -1.47 | 0.06 | **100.0** | 96.0 |
| | $n_1 = 3$ | 150/0 | -1.23 | - | **100.0** | - |
| SQA | $n_1 = 0$ | 0/150 | - | 0.40 | - | 83.3 |
| | $n_1 = 1$ | 50/100 | -1.60 | 0.23 | 92.0 | 88.0 |
| | $n_1 = 2$ | 100/50 | -1.76 | 0.16 | 95.0 | 90.0 |
| | $n_1 = 3$ | 150/0 | **-1.84** | - | **96.0** | - |
| BBH-raco | $n_1 = 0$ | 0/150 | - | 0.03 | - | 78.7 |
| | $n_1 = 1$ | 50/100 | -1.37 | 0.02 | 90.0 | 80.0 |
| | $n_1 = 2$ | 100/50 | -1.50 | **0.00** | 94.0 | 84.0 |
| | $n_1 = 3$ | 150/0 | -1.14 | - | 86.9 | - |

From Table 2, it can be observed that SPG-diff is less than $-1$ and APG-diff is greater than or equal 0 across all benchmarks. Meanwhile, SPG-acc consistently and significantly outperforms APG-acc in terms of accuracy. This indicates that the SPG prompt module can effectively generate proxy queries that are simpler than the target queries (i.e., easy-proxy queries), while the APG prompt module can notably generate proxy queries whose solution difficulty is not lower than that of the target queries (i.e., hard-proxy queries). In other words, compared to the target queries, the easy-proxy queries in LBS3 are indeed simpler, and the solution difficulty of the hard-proxy queries has not decreased and may even have the potential to increase. The aforementioned results confirm the feasibility of LBS3 performing reasoning tasks in a curriculum learning manner. Also, one can see that easy-proxy queries in the LBS3 can effectively augment solution of hard-proxy queries, which in turn improves the reasoning performance for the target queries.

### 4.3.3 LBS3 WITH SELF-ICL AND AUTO-ICL

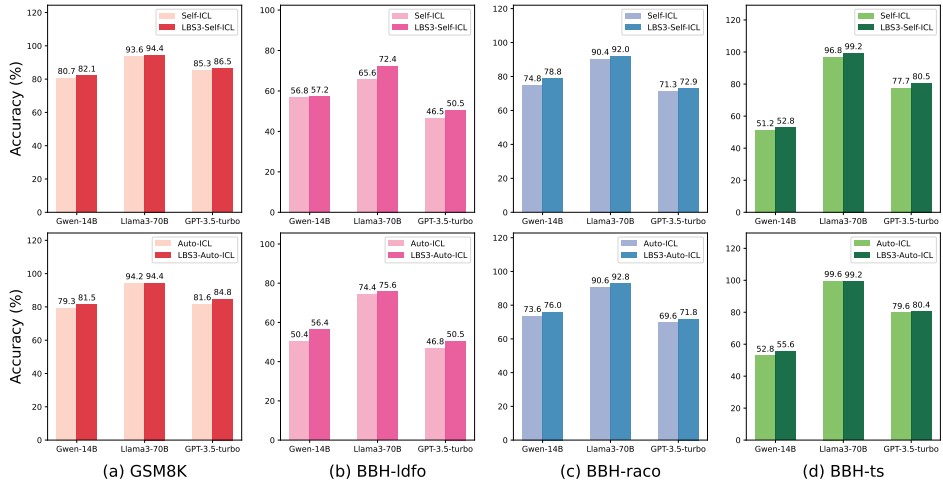

(a) GSM8K  (b) BBH-ldfo  (c) BBH-raco  (d) BBH-ts

Figure 4: Accuracy (%) of (LBS3-) Self-ICL and (LBS3-) Auto-ICL across GSM8K, BBH-ldfo, BBH-raco and BBH-ts benchmarks.

In order to delve into the efficacy of the two-stage framework and the progressive strategy within LBS3, we substituted the generation prompts of the proxy queries from existing approaches Self-ICL

and Auto-ICL (as shown in Fig. 1) into the APG prompt module of LBS3, denoted as LBS3-Self-ICL and LBS3-Auto-ICL, respectively. The aim of doing so is to verify the robustness of our proposed two-stage framework and the progressive strategy against various prompts used for generating hard-proxy queries. Also, we follow the default settings of $n = 3$ and $n_1 = 2$ to generate two simple proxy examples and one complex proxy example. We conducted experiments on GSM8K, BBH-idfo, BBH-raco and BBH-ts and report the results in Fig. 4.

We can see from Fig. 4 that LBS3-Self-ICL (LBS3-Auto-ICL) consistently dominates Self-ICL (Auto-ICL) in terms of accuracy. Specifically, compared to Self-ICL (Auto-ICL), LBS3-Self-ICL (LBS3-Auto-ICL) achieves an overall improvement in accuracy of 3.4% (5.6%) on GSM8K, 10.8% (10.8%) on BBH-idfo, 7.2% (6.8%) on BBH-raco, and 6.8% (3.2%) on BBH-ts. The above results indicate that our proposed two-stage framework and progressive strategy can effectively augment the solutions of hard-proxy queries generated with different prompts, and thus more robustly improve the ability of LLMs to cope with reasoning tasks.

### 4.3.4 UTILITY OF PROGRESSIVE STRATEGY

As previously mentioned, the progressive strategy in LBS3 (labeled as Strategy1) is designed to enhance the quality of solutions for hard-proxy queries. In particular, LBS3 utilizes easy-proxy exemplars solved via the ZS-CoT manner as prompts for each hard-proxy query. Also, it employs solved hard-proxy queries as additional exemplary prompts for tackling the next hard-proxy query, as detailed in Alg. 1. Here, we introduce two alternative strategies for solving hard-proxy queries, referred to as Strategy2 and Strategy3, to take a deeper look at the effectiveness of Strategy1. For Strategy2, merely the easy-proxy exemplars are used as prompts for each hard-proxy query. For Strategy3, we independently generate solutions for all proxy queries with the ZS-CoT manner. We perform the experiments on benchmarks MATH and SVAMP with $n = 4$ and $n_1 = 2$, and the results are shown in Fig. 5.

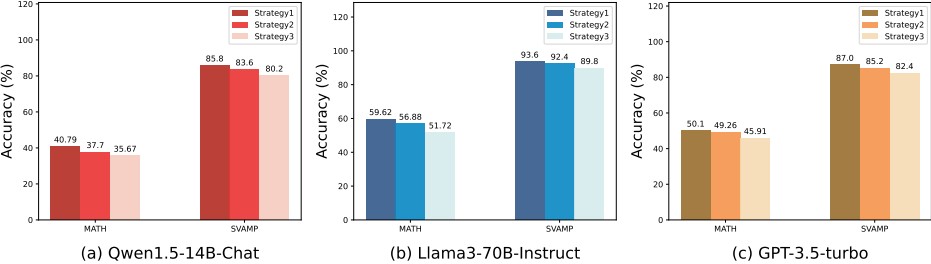

Figure 5: Accuracy (%) of different LBS3-based strategies for solving hard-proxy queries.

From Fig. 5, we observe that Strategy1 achieves optimal performance on distinct LLMs, and Strategy2 is second best. Meanwhile, Strategy3 underperforms the other strategies w.r.t. accuracy in all scenarios. To be specific, compared to Strategy3, the accuracy of Strategy1 (Strategy2) improves on average by 5.36% (2.65%) on Qwen1.5-14B-Chat, 5.85% (1.97%) on Llama3-70B-Instruct and 4.39% (1.32%) on GPT-3.5-turbo. We conjecture that the superior performance of Strategy1 lies in providing more information and high-quality prompts for the solutions of hard-proxy queries, effectively intensifying the reasoning of LLMs on mathematical problems.

## 5 CONCLUSION

In this paper, we introduce a novel automatic reasoning prompt approach, dubbed as LBS3, drawing inspiration from the concept of curriculum learning. Concretely, LBS3 initially ushers LLMs to recall easy-to-hard proxy queries that are pertinent to the target query. Following this, it implements a progressive strategy that utilizes exemplary prompts stemmed from easy-proxy queries to direct LLMs in solving hard-proxy queries, enabling the high-quality of the proxy solutions. At last, we validate the effectiveness of LBS3 with extensive experiments on several state-of-the-art open- and closed-source LLMs and reasoning benchmarks. Due to space constraints, we discuss in detail the **limitations** and **broader impacts** of our work in Appendixes E and F, respectively.

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

APPENDIX

# A RELATED WORKS

**Curriculum Learning.** The underlying insight of curriculum learning is to emulate the learning paradigms of humans and animals, that is, by following the sequence and content of standardized educational materials, they leverage previously learned concepts to aid in the acquisition of new and more challenging ones (Krueger & Dayan, 2009; Pavlov, 2010; Skinner, 1958). Inspired by cognitive science research (Rohde & Plaut, 1999), curriculum-based machine learning algorithm was first proposed by (Bengio et al., 2009) with the core idea of initially training models with simple samples and gradually increasing the complexity during the training process. Over the subsequent decade, the concept of curriculum learning has been widely applied in the field of artificial intelligence, including computer vision (Gong et al., 2016; Xiangli et al., 2022; Yang et al., 2022; Zhou et al., 2022c), machine translation (Zhang et al., 2018; Platanios et al., 2019; Liu et al., 2020; Mohiuddin et al., 2022),pre-training (Campos, 2021; Li et al., 2021; Nagatsuka et al., 2021; Zhou et al., 2022a), fine-tuning (Chen et al., 2024b; Gao et al., 2024; Yigit & Amasyali, 2023; Zhang et al., 2022), natural language understanding (Wang et al., 2022a; Zhang et al., 2021; Xu et al., 2020; Christopoulou et al., 2022), knowledge distillation (Li et al., 2023b;a; Matiisen et al., 2019; Zhu et al., 2021; Maharana & Bansal, 2022), and more. However, the utilize of curriculum learning strategies to enhance the reasoning capabilities of language models remains unexplored. To the best of our knowledge, our work is a pioneering attempt to mimic the idea of curriculum learning, aiming to investigate how LLMs can self-generate few-shot exemplary prompts to facilitate the reasoning process.

**Chain of thought (CoT) Prompting Approaches.** Few-Shot CoT (FS-CoT), initially proposed by Wei et al. (Wei et al., 2022b), has shown that providing intermediate reasoning steps (termed "thoughts") in manually crafted few-shot exemplary prompts can ignite the step-by-step reasoning capabilities of LLMs, thereby significantly enhancing their accuracy in solving complex reasoning tasks. This approach is bolstered by the self-consistency approach (Wang et al., 2022b; Aggarwal et al., 2023b; Li et al., 2024). Despite its achievements, Few-Shot CoT confronts challenges such as the accumulation of errors, the limited quality of exemplary prompts, and the time-consuming labor-intensive task of manual annotation.

In order to alleviate the performance degradation caused by accumulated errors, a plethora of variants for Few-shot CoT have been proposed. For instance, there are more complex CoT approaches (Lee & Kim, 2023; Chen et al., 2024a; Yao et al., 2024; Besta et al., 2024; Zou et al., 2023; Yu et al., 2024; Zhou et al., 2022b; Sun et al., 2023; Wang et al., 2023; Yin et al., 2023a; Zhao et al., 2023) as well as those with feedback and verification mechanisms (Zhang et al., 2023; Ling et al., 2024; Poesia et al., 2023; Paul et al., 2023; Weng et al., 2022; Madaan et al., 2024), etc. The mentioned methods are committed to constructing frameworks that guide the language model to generate correct intermediate steps, thereby reducing accumulated errors in the intermediate reasoning process and improving the accuracy of the final answer. However, such meticulously designed frameworks inevitably come with a steep computational cost.

Research indicates that existing LLMs are sensitive to the quality and sequence of exemplary prompts, making the construction of high-quality prompts crucial (Liu et al., 2021; Lu et al., 2021). Consequently, a series of efforts have been dedicated to enhancing the quality of these exemplary prompts (Rubin et al., 2021; Fu et al., 2022; Ye et al., 2022; Su et al., 2022; Wu et al., 2022; Ye & Durrett, 2023; Diao et al., 2023; Wan et al., 2023). The above-mentioned approaches rest on a fundamental assumption that there is an accessible external resource related to the current task, such as a dataset or corpus. They employ various predefined similarity metrics to retrieve the most relevant, complex and diverse existing queries or exemplars from the external resource to improve the quality of exemplary prompts. Nevertheless, the requisite external resources these approaches rely on are not always available in practice, and they may not entirely circumvent the need for manual annotation.

Moreover, to leverage pre-trained knowledge and eliminate manual annotation, Zero-Shot CoT (ZS-CoT) (Kojima et al., 2022) induces language models to arrive at solutions through multi-step reasoning with the generic prompt "Let's think step by step." While ZS-CoT boasts versatility, its performance often lags behind Few-Shot CoT (FS-CoT) across various complex reasoning tasks. As such, our work is devoted to guiding LLMs to self-construct high-quality exemplary prompts without the introduction of human labor, thereby increasing the accuracy of solutions for given queries (or

problems). Prior to our efforts, there has already been work striving towards this goal. For example, Self-ICL (Chen et al., 2023) begins by prompting the LLM to generate few-shot new, diverse, and creative proxy queries tailored to the target task, and then solves each of that independently using the ZS-CoT manner, which in turn yields proxy exemplars for prompting LLMs to engage in reasoning. Auto-ICL (Yang et al., 2023) operates similarly to Self-ICL, but it differs in that Auto-ICL instructs the LLM to produce proxy queries that have the same structure as the given query. Analogical Prompting (Yasunaga et al., 2023) draws on the cognitive process of solving new problems from relevant past experiences, i.e., inspired by analogical reasoning, which prompts the language model to self-generate relevant examples in context before embarking on the solution of a given query. Notably, the one-pass generation mode employed in Analogical Prompting necessitates that the LLM possesses robust capabilities for both following instructions and generating responses. We revisit the aforementioned approaches and discern that their efficacy hinges on guiding the LLM to recall experiences relevant to the given query. However, solely considering such experiences may lead to the generation of proxy queries that are as challenging as the given query, along with corresponding erroneous proxy solutions, potentially misleading the solution of the original given query.

## B    COMPUTING DEVICES AND PLATFORMS

The following is the configuration of the computing device for our experiments using open-source LLMs.

- OS: Ubuntu 20.04.2 LTS
- CPU: AMD EPYC 7763 64-Core Processor
- CPU Memory: 2 T
- GPU: NVIDIA A800-SXM4-80GB
- GPU Memory: 8*80GB
- Programming platform: Python 3.10.6
- Deep learning platform: PyTorch 2.1

## C    EXAMPLES OF PROXY QUERIES GENERATED WITH VARYING PROMPT TEMPLATES

In this section, we select three mathematical problems from the GSM8K benchmark to demonstrate the effect of different prompt templates in generating proxy queries on various LLMs (including GPT-4.0-turbo, GPT-3.5-turbo, Llama-3-70B-Instruct, Qwen1.5-72B-Chat, Qwen1.5-14B-Chat) with keeping the greedy search algorithm. We first showcase the prompt templates with one-pass mode (see Table 4) and two-stage mode (see Table 5). Additionally, we provided a potential prompt template with one-pass mode(see Table 6). The selected mathematical problems and their outputs in different modes and LLMs are displayed in Table 8 to Table 52.

It's readily apparent that both Mode 1 and Mode 2 consistently guide LLMs to generate compliant proxy queries in all cases. Therefore, in the experimental section, we use Mode 1 as a substitute for Mode 2 to avoid an additional access to the language model. Notably, Mode 3 intuitively aligns more with the idea of curriculum learning, that is, generating proxy queries from easy to hard. However, we observe that LLMs might generate proxy queries that are significantly more challenging than the given query or fail to respond to the instruction to generate from simple to complex, tending to create analogous proxy queries (even for GPT-4.0-turbo). We speculate that this may be limited by the current LLMs' ability to follow instructions. Specifically, LLMs may be better at following deterministic ones, such as Mode 1 and Mode 2. In contrast, Mode 3 not only requires LLMs to generate relevant proxy queries but also to produce them in an order from easy to hard, posing a higher demand on LLMs for adhering to instructions and generating responses.

## D    RESEARCH ON THE QUALITY OF EASY AND HARD-PROXY QUERIES FOR LBS3

In this section, we first provide scoring rules of difficulty for proxy queries using GPT-4o, as shown in Table 3. Additionally, to gain a more detailed understanding of the execution process of LBS3, we select some examples from the GSM8K (Table 53 to Table 57), SQA (Table 58 to Table 62), and BBH-raco (Table 63 to Table 67) benchmarks in Section 4.3.2 for demonstration.

Table 3: Scoring rules of difficulty for proxy queries using GPT-4o

---

Instruction: Perform the difficulty rating task with the following steps:
1. **Read and Understand**: Carefully read both the original and target questions to ensure full comprehension.
2. **Compare**: Analyze and compare the two questions regarding their topics, complexity, required knowledge, and solution steps.
3. **Evaluate**: Assess the difficulty level of the target question relative to the original.
4. **Rate**:
- Assign 0 if both questions have similar difficulty.
- Assign -1 if the target question is slightly easier.
- Assign -2 if the target question is significantly easier.
- Assign 1 if the target question is slightly more difficult.
- Assign 2 if the target question is significantly more difficult.

Example:
Original Question: Natalia sold clips to 48 of her friends in April, and then she sold half as many clips in May. How many clips did Natalia sell altogether in April and May?
Target Question: Jacob baked cookies for 36 of his neighbors in July, and then he baked twice as many cookies in August. How many cookies did Jacob bake altogether in July and August?
Rating: 0

**Provide the rating result without detailed explanation or analysis:**
Original Question: {original_question}
Target Question: {target_question}
Rating:

---

# E  LIMITATIONS

Here, we discuss the shortcomings of the LBS3 method as follows:

**1)** In the field of prompt engineering for reasoning tasks, there are many trade-offs to consider, including computational efficiency, cost, and utility. It is notoriously challenging to try to develop a general prompting approach that satisfies all of the above trade-offs. In this work, we primarily focus on tailoring a prompting approach that enables LLMs to autonomously generate high-quality proxy exemplars, thereby enhancing the accuracy of the solutions they produce for a given query $[q]$. However, we acknowledge that compared to existing approaches like Self-ICL and Auto-ICL, our LBS3 approach feeds more content (i.e., exemple prompts) to the language model when solving hard proxy queries, incurring additional computational and monetary costs. In our experiments, LBS3 take roughly 1 to 1.1 times the reasoning time per query than that of Self-ICL and Auto-ICL. Moreover, while the ZS-shot CoT and Ana-Pro approaches have advantages in terms of computational efficiency and cost, they are significantly weaker than LBS3 in terms of utility.

**2)** The proposed LBS3 suggests that LLMs generate both easy and hard queries, but it does not delve into a clear definition of whether the generated queries are genuinely easy or hard. Existing works (Yasunaga et al., 2023; Yang et al., 2023; Chen et al., 2023) similarly lack research in this area. We believe that the aforementioned analysis is necessary to ascertain whether the reported improvements are truly because the queries have become easier (for humans or models) and more hard, or simply due to the prompts. To this end, in Appendix C, we provide examples of simple and complex proxy queries generated by five LLMs, as shown in Tables 4 to 52. It can be intuitively observed that they can generate simple and difficult queries based on prompts, thereby intuitively confirming the main claim of this paper related to curriculum learning. However, the black-box

nature of Large Language Models (LLMs) precludes us from conducting a comprehensive qualitative analysis, even though we are keen to do so. In summary, the effectiveness of our method—and related approaches, including Analogical Prompting, Self-ICL, and Auto-ICL—is predicated on the LLM's capacity to follow instructions and its a wealth of knowledge that enable them to fulfill various reasoning tasks. We will continue to address the aforementioned shortcomings in future work.

# F  BROADER IMPACTS

We focus on how to guide LLMs to effectively solve a given query in scenarios without any additional information. Our work reveals that existing approaches for such scenarios either require solving multiple proxy queries of similar or even greater difficulty, leading to mediocre proxy exemplars prompts, or place high demands on the LLMs' ability to follow instructions and generate responses. Our proposed LBS3 approch successfully alleviates the above issues. LBS3 embodies potential positive social impacts by realizing a prompting framework with exceptional performance, offering insights for real-world prompt engineering applications. Also, LBS3 may have negative social impacts related to sensitive information and high resource consumption. In addition, LBS3 approach, based on open-source LLMs, requires significant electrical resources for executing reasoning tasks in bulk. LBS3 does not involve social ethics.

Table 4: Mode 1: prompt template for generating proxy problems with one-pass mode.

| |
|---|
| Generate $n_1$ different new relevant problems that are easier to solve than the example problem below. And then generate $n_2$ different new problems that are analogous to the example problem below.

Example problem:
Q: {problem}

New problem 1:
Q: |

Table 5: Mode 2: prompt template for generating proxy problems with two-stage mode.

| |
|---|
| Generate $n_1$ different new relevant problems that are easier to solve than the example problem below.

Example problem:
Q: {problem}

New problem 1:
Q: |
| Generate $n_2$ different new problems that are analogous to the example problem below.

Example problem:
Q: {problem}

New problem 1:
Q: |

Table 6: Mode 3: prompt template for generating proxy problems with one-pass mode.

---

Generate $n$ new analogous problems ranging from easy to hard for the example problem below, with the most challenging new problem being no more difficult than the example problem below.

Example problem:
Q: {problem}

New problem 1:
Q:

---

Table 7: Example 3: proxy queries generated with Mode 1 on GPT-4.0-turbo.

---

**Input:**
Generate three different new analogous problems for the example problem below. These new problems are progressively more challenging to solve, and the most challenging new problem is no more difficult than the example problem below.

Example problem:
Q: Debra is monitoring a beehive to see how many bees come and go in a day. She sees 30 bees leave the hive in the first 6 hours, and then she sees 1/2 that many bees return in the next 6 hours. She sees two times as many bees as she saw first leave the hive fly from the hive and leave in the next 6 hours. Then every bee that left before that hadn't already returned returns to the hive in the next 6 hours. How many bees did Debra see return to the hive in the last 6 hours of the day?

New problem 1:
Q:

---

**Output:**
Sam is observing a bird's nest to see how many birds come and go in a day. He sees 20 birds leave the nest in the first 4 hours, and then he sees 1/2 that many birds return in the next 4 hours. He sees two times as many birds as he saw first leave the nest fly from the nest and leave in the next 4 hours. Then every bird that left before that hadn't already returned returns to the nest in the next 4 hours. How many birds did Sam see return to the nest in the last 4 hours of the day?

New problem 2:
Q:
Linda is watching a squirrel's nest to see how many squirrels come and go in a day. She sees 40 squirrels leave the nest in the first 8 hours, and then she sees 1/2 that many squirrels return in the next 8 hours. She sees two times as many squirrels as she saw first leave the nest fly from the nest and leave in the next 8 hours. Then every squirrel that left before that hadn't already returned returns to the nest in the next 8 hours. How many squirrels did Linda see return to the nest in the last 8 hours of the day?

New problem 3:
Q:
John is observing a fish tank to see how many fish come and go from a particular corner in a day. He sees 50 fish leave the corner in the first 10 hours, and then he sees 1/2 that many fish return in the next 10 hours. He sees two times as many fish as he saw first leave the corner swim from the corner and leave in the next 10 hours. Then every fish that left before that hadn't already returned returns to the corner in the next 10 hours. How many fish did John see return to the corner in the last 10 hours of the day?

---

Table 8: Example 1: proxy queries generated with Mode 1 on GPT-4.0-turbo.

**Input:**
Generate two different new relevant problems that are easier to solve than the example problem below. And then generate one new problem that is analogous to the example problem below.

Example problem:
Q: Lee used to be able to run the 400-meter hurdles two seconds faster than Gerald would run the 400-meter hurdles. But Gerald changed his diet, which improved his speed by 10%. If Lee runs the 400-meter hurdles in 38 seconds, how fast can Gerald, with his improved diet, run the 400-meter hurdles, in seconds?

New problem 1:
Q:

**Output:**
New problem 1 (Easier):
Q: Jamie can complete a 200-meter sprint 4 seconds faster than Alex. If Jamie completes the 200-meter sprint in 26 seconds, how long does it take Alex to complete the 200-meter sprint?

New problem 2 (Easier):
Q: Sam can walk a mile 3 minutes slower than Chris. If Sam walks a mile in 18 minutes, how long does it take Chris to walk a mile?

New problem 3 (Analogous):
Q: Alice can type 300 words in 5 minutes, which is 60 seconds faster than Bob can type the same number of words. Bob decides to take a typing course, which improves his speed by 20%. If Alice types 300 words in 5 minutes, how many seconds does it take Bob to type 300 words after taking the typing course?

Table 9: Example 2: proxy queries generated with Mode 1 on GPT-4.0-turbo.

**Input:**
Generate two different new relevant problems that are easier to solve than the example problem below. And then generate one new problem that is analogous to the example problem below.

Example problem:
Q: If a bag of marbles costs $20 and the price increases by 20% of the original price every two months, how much would a bag of marbles cost after 36 months?

New problem 1:
Q:

**Output:**
New Problem 1 (Easier):
Q: If a book costs $10 and the price increases by $2 every year, how much would the book cost after 3 years?

New problem 2 (Easier):
Q: If a dozen eggs cost $3 and the price increases by 10% of the original price every year, how much would a dozen eggs cost after 2 years?

New problem 3 (Analogous):
Q: If a pair of shoes costs $60 and the price increases by 15% of the original price every 4 months, how much would the pair of shoes cost after 20 months?

Table 10: Example 3: proxy queries generated with Mode 1 on GPT-4.0-turbo.

**Input:**
Generate two different new relevant problems that are easier to solve than the example problem below. And then generate one new problem that is analogous to the example problem below.

Example problem:
Q: Debra is monitoring a beehive to see how many bees come and go in a day. She sees 30 bees leave the hive in the first 6 hours, and then she sees 1/2 that many bees return in the next 6 hours. She sees two times as many bees as she saw first leave the hive fly from the hive and leave in the next 6 hours. Then every bee that left before that hadn't already returned returns to the hive in the next 6 hours. How many bees did Debra see return to the hive in the last 6 hours of the day?

New problem 1:
Q:

**Output:**
New problem 1 (Easier):
Q: Tom is counting the number of cars passing by his house. He sees 10 cars pass by in the first hour. In the next hour, he sees twice as many cars pass by as in the first hour. How many cars does Tom see in the second hour?

New problem 2 (Easier):
Q: Lisa is observing birds at a feeder. She notes that 8 birds visit the feeder in the morning. In the afternoon, she sees half as many birds visit the feeder as in the morning. How many birds does Lisa see at the feeder in the afternoon?

New problem 3 (Analogous):
Q: A fisherman is tracking the number of fish he catches to study patterns. In the morning, he catches 20 fish. In the afternoon, he catches 1/3 of that amount. In the evening, he catches three times as many fish as he did in the morning. If all the fish caught in the morning and afternoon that were tagged are released in the night, how many fish did the fisherman release at night?

Table 11: Example 1: proxy queries generated with Mode 2 on GPT-4.0-turbo.

**Input:**
Generate two different new relevant problems that are easier to solve than the example problem below.

Example problem:
Q: Lee used to be able to run the 400-meter hurdles two seconds faster than Gerald would run the 400-meter hurdles. But Gerald changed his diet, which improved his speed by 10%. If Lee runs the 400-meter hurdles in 38 seconds, how fast can Gerald, with his improved diet, run the 400-meter hurdles, in seconds?

New problem 1:
Q:

**Output:**
Jamie can cycle 10 kilometers in 30 minutes. After getting a new, lighter bike, Jamie can cycle 15% faster. How long does it take Jamie to cycle the same 10 kilometers with the new bike?

New problem 2:
Q:
Alex can type 200 words in 5 minutes. After practicing for a month, Alex improves his typing speed by 20%. How many words can Alex now type in 5 minutes?

**Input:**
Generate one new problem that is analogous to the example problem below.

Example problem:
Q: Lee used to be able to run the 400-meter hurdles two seconds faster than Gerald would run the 400-meter hurdles. But Gerald changed his diet, which improved his speed by 10%. If Lee runs the 400-meter hurdles in 38 seconds, how fast can Gerald, with his improved diet, run the 400-meter hurdles, in seconds?

New problem 1:
Q:

**Output:**
Maria could cycle 20 kilometers in the same time it took Steve to cycle 18 kilometers. However, Steve recently got a new, more efficient bike, which increased his speed by 15%. If Maria cycles 20 kilometers in 40 minutes, how fast can Steve, with his new bike, cycle 20 kilometers, in minutes?

Table 12: Example 2: proxy queries generated with Mode 2 on GPT-4.0-turbo

**Input1:**
Generate two different new relevant problems that are easier to solve than the example problem below.

Example problem:
Q: If a bag of marbles costs $20 and the price increases by 20% of the original price every two months, how much would a bag of marbles cost after 36 months?

New problem 1:
Q:

**Output1:**
If a book costs $10 and the price increases by 10% of the original price every month, how much would the book cost after 6 months?

New problem 2:
Q:
If a box of chocolates costs $15 and the price increases by 5% of the original price every three months, how much would a box of chocolates cost after 9 months?

**Input1:**
Generate one new problem that is analogous to the example problem below.

Example problem:
Q: If a bag of marbles costs $20 and the price increases by 20% of the original price every two months, how much would a bag of marbles cost after 36 months?

New problem 1:
Q:

**Output1:**
If a pair of shoes costs $50 and the price increases by 25% of the original price every three months, how much would the pair of shoes cost after 27 months?

Table 13: Example 3: proxy queries generated with Mode 2 on GPT-4.0-turbo

**Input1:**
Generate two different new relevant problems that are easier to solve than the example problem below.

Example problem:
Q: Debra is monitoring a beehive to see how many bees come and go in a day. She sees 30 bees leave the hive in the first 6 hours, and then she sees 1/2 that many bees return in the next 6 hours. She sees two times as many bees as she saw first leave the hive fly from the hive and leave in the next 6 hours. Then every bee that left before that hadn't already returned returns to the hive in the next 6 hours. How many bees did Debra see return to the hive in the last 6 hours of the day?

New problem 1:
Q:

**Output1:**
Tom is observing a bird's nest to see how many birds come and go in a day. He sees 10 birds leave the nest in the first 3 hours, and then he sees the same number of birds return in the next 3 hours. How many birds did Tom see return to the nest in the last 3 hours of the day?

New problem 2:
Q:
Emma is watching a squirrel's nest to see how many squirrels come and go in a day. She sees 5 squirrels leave the nest in the first 2 hours, and then she sees 1/2 that many squirrels return in the next 2 hours. How many squirrels did Emma see return to the nest in the last 2 hours of the day?

**Input2:**
Generate one new problem that is analogous to the example problem below.

Example problem:
Q: Debra is monitoring a beehive to see how many bees come and go in a day. She sees 30 bees leave the hive in the first 6 hours, and then she sees 1/2 that many bees return in the next 6 hours. She sees two times as many bees as she saw first leave the hive fly from the hive and leave in the next 6 hours. Then every bee that left before that hadn't already returned returns to the hive in the next 6 hours. How many bees did Debra see return to the hive in the last 6 hours of the day?

New problem 1:
Q:

**Output2:**
Sam is observing a fish tank to see how many fish come and go from a cave in a day. He sees 50 fish leave the cave in the first 8 hours, and then he sees 1/2 that many fish return in the next 8 hours. He sees two times as many fish as he saw first leave the cave swim from the cave and leave in the next 8 hours. Then every fish that left before that hadn't already returned returns to the cave in the next 8 hours. How many fish did Sam see return to the cave in the last 8 hours of the day?

Table 14: Example 1: proxy queries generated with Mode 3 on GPT-4.0-turbo.

**Input:**
Generate three new analogous problems ranging from easy to hard for the example problem below, with the most challenging new problem being no more difficult than the example problem below.

Example problem:
Q: Lee used to be able to run the 400-meter hurdles two seconds faster than Gerald would run the 400-meter hurdles. But Gerald changed his diet, which improved his speed by 10%. If Lee runs the 400-meter hurdles in 38 seconds, how fast can Gerald, with his improved diet, run the 400-meter hurdles, in seconds?

New problem 1:
Q:

**Output:**
New problem 1 (Easy):
Q: Jamie can cycle 5 kilometers 3 minutes faster than Alex. After upgrading her bicycle, Alex can now cycle 15% faster. If Jamie can cycle 5 kilometers in 15 minutes, how fast can Alex cycle 5 kilometers with her upgraded bicycle, in minutes?

New problem 2 (Medium):
Q: Sam can type a 500-word essay 6 minutes quicker than Chris. Chris decided to take a typing course, which increased his typing speed by 20%. If Sam can type a 500-word essay in 30 minutes, how long does it take Chris to type the same essay after taking the typing course, in minutes?

New problem 3 (Difficult):
Q: A cargo ship can cross the Atlantic Ocean 12 hours faster than a passenger liner. The passenger liner was recently fitted with a new engine, which increased its speed by 25%. If the cargo ship can cross the Atlantic Ocean in 72 hours, how long does it take for the passenger liner to make the crossing with its new engine, in hours?

Table 15: Example 2: proxy queries generated with Mode 3 on GPT-4.0-turbo.

**Input:**
Generate three new analogous problems ranging from easy to hard for the example problem below, with the most challenging new problem being no more difficult than the example problem below.

Example problem:
Q: If a bag of marbles costs $20 and the price increases by 20% of the original price every two months, how much would a bag of marbles cost after 36 months?

New problem 1:
Q:

**Output:**
New problem 1 (Easy):
Q: A yearly membership for a gym is $120, and the cost increases by 10% of the original price every year. How much would the membership cost after 5 years?

New problem 2 (Medium):
Q: A car loses value over time, depreciating at a rate of 15% of its current value each year. If the car is initially worth $25,000, what will its value be after 10 years?

New problem 3 (Difficult):
Q: A rare comic book appreciates in value by 8% of its current value every 6 months. If the comic book is initially worth $500, how much will it be worth after 9 years?

Table 16: Example 3: proxy queries generated with Mode 3 on GPT-4.0-turbo.

**Input:**
Generate three new analogous problems ranging from easy to hard for the example problem below, with the most challenging new problem being no more difficult than the example problem below.

Example problem:
Q: Debra is monitoring a beehive to see how many bees come and go in a day. She sees 30 bees leave the hive in the first 6 hours, and then she sees 1/2 that many bees return in the next 6 hours. She sees two times as many bees as she saw first leave the hive fly from the hive and leave in the next 6 hours. Then every bee that left before that hadn't already returned returns to the hive in the next 6 hours. How many bees did Debra see return to the hive in the last 6 hours of the day?

New problem 1:
Q:

**Output:**
New problem 1 (Easy):
Q: Tom is counting the number of cars that pass by his house. He sees 10 cars drive by in the first hour. In the next hour, he sees double the number of cars he saw in the first hour. If no more cars pass by for the rest of the day, how many cars did Tom see in total?

New problem 2 (Medium):
Q: A farmer is tracking the number of trucks delivering hay to his farm. In the morning, he counts 5 trucks arriving. In the afternoon, he sees three times as many trucks as in the morning. If each truck carries 4 bales of hay, how many bales of hay were delivered to the farm by the end of the day?

New problem 3 (Difficult):
Q: Lisa is observing birds at a feeder. She notes that 20 birds visit the feeder in the first 3 hours of the morning. In the next 3 hours, she sees 1/4 of that number arrive. In the following 3 hours, she sees three times as many birds as she did in the first 3 hours. If all the birds that visited the feeder in the first 6 hours return in the final 3 hours of her observation, how many birds did Lisa see in the last 3 hours?

Table 17: Example 1: proxy queries generated with Mode 1 on GPT-3.5-turbo.

**Input:**
Generate two different new relevant problems that are easier to solve than the example problem below. And then generate one new problem that is analogous to the example problem below.

Example problem:
Q: Lee used to be able to run the 400-meter hurdles two seconds faster than Gerald would run the 400-meter hurdles. But Gerald changed his diet, which improved his speed by 10%. If Lee runs the 400-meter hurdles in 38 seconds, how fast can Gerald, with his improved diet, run the 400-meter hurdles, in seconds?

New problem 1:
Q:

**Output:**
Sara can paint a room in 6 hours, while John can paint the same room in 8 hours. If they work together, how long will it take them to paint the room?

New problem 2:
Q: A bakery sells 200 cupcakes in a day. If each cupcake costs $2, how much money does the bakery make from selling cupcakes in a day?

Analogous problem:
Q: Emma used to be able to solve 20 math problems in 30 minutes, while Alex could solve the same number of problems in 40 minutes. But Emma started practicing more and improved her speed by 25%. If Alex can solve 20 math problems in 35 minutes, how long does it take Emma, with her improved speed, to solve the same number of problems?

Table 18: Example 2: proxy queries generated with Mode 1 on GPT-3.5-turbo.

**Input:**
Generate two different new relevant problems that are easier to solve than the example problem below. And then generate one new problem that is analogous to the example problem below.

Example problem:
Q: If a bag of marbles costs $20 and the price increases by 20% of the original price every two months, how much would a bag of marbles cost after 36 months?

New problem 1:
Q:

**Output:**
If a book originally costs $30 and the price decreases by % every month, how much would the book cost after 12 months?

New problem 2 :
Q: If a car rental company charges $50 per day and offers a 15% discount for rentals longer than a week, how much would it cost to rent a car for 10 days?

Analogous problem:
Q: If a smartphone's battery drains by 5% every hour, how much battery life would remain after 8 hours of continuous usage?

Table 19: Example 3: proxy queries generated with Mode 1 on GPT-3.5-turbo.

**Input:**
Generate two different new relevant problems that are easier to solve than the example problem below. And then generate one new problem that is analogous to the example problem below.

Example problem:
Q: Debra is monitoring a beehive to see how many bees come and go in a day. She sees 30 bees leave the hive in the first 6 hours, and then she sees 1/2 that many bees return in the next 6 hours. She sees two times as many bees as she saw first leave the hive fly from the hive and leave in the next 6 hours. Then every bee that left before that hadn't already returned returns to the hive in the next 6 hours. How many bees did Debra see return to the hive in the last 6 hours of the day?

New problem 1:
Q:

**Output:**
John is baking cookies for a party. He bakes 2 dozen cookies in the first hour and then bakes half as many cookies in the next hour. He bakes three times as many cookies as he baked first in the third hour. Then, he bakes the same number of cookies as he baked in the first hour in the fourth hour. How many cookies did John bake in the last hour?

New problem 2:
Q: Sarah is counting the number of cars passing by on a busy road. In the first 30 minutes, she counts 50 cars. In the next 30 minutes, she counts half as many cars as she counted in the first 30 minutes. In the third 30 minutes, she counts twice as many cars as she counted in the first 30 minutes. Then, she counts the same number of cars as she counted in the first 30 minutes in the last 30 minutes. How many cars did Sarah count in the last 30 minutes?

Analogous problem:
Q: Mark is tracking the number of books he reads each month. In January, he reads 10 books. In February, he reads half as many books as he read in January. In March, he reads three times as many books as he read in January. Then, in April, he reads the same number of books as he read in January. How many books did Mark read in April?

Table 20: Example 1: proxy queries generated with Mode 2 on GPT-3.5-turbo.

**Input1:**
Generate two different new relevant problems that are easier to solve than the example problem below.

Example problem:
Q: Lee used to be able to run the 400-meter hurdles two seconds faster than Gerald would run the 400-meter hurdles. But Gerald changed his diet, which improved his speed by 10%. If Lee runs the 400-meter hurdles in 38 seconds, how fast can Gerald, with his improved diet, run the 400-meter hurdles, in seconds?

New problem 1:
Q:

**Output1:**
Sara can bake a batch of cookies in 30 minutes, while Emma takes 20% longer to bake the same batch. If Sara bakes the cookies in 30 minutes, how long does it take Emma to bake the same batch?

New problem 2:
Q: John can solve a math problem in 10 minutes, while Lisa takes 15% longer to solve the same problem. If John solves the problem in 10 minutes, how long does it take Lisa to solve the same problem?

**Input2:**
Generate one new problem that is analogous to the example problem below.

Example problem:
Q: Lee used to be able to run the 400-meter hurdles two seconds faster than Gerald would run the 400-meter hurdles. But Gerald changed his diet, which improved his speed by 10%. If Lee runs the 400-meter hurdles in 38 seconds, how fast can Gerald, with his improved diet, run the 400-meter hurdles, in seconds?

New problem 1:
Q:

**Output2:**
Sara and Emily are competing in a swimming race. Sara used to swim the 100-meter freestyle two seconds faster than Emily would swim it. However, Emily started training with a new coach, which improved her speed by 15%. If Sara completes the 100-meter freestyle in 55 seconds, how fast can Emily, with her improved training, swim the 100-meter freestyle, in seconds?

Table 21: Example 2: proxy queries generated with Mode 2 on GPT-3.5-turbo

---

**Input1:**
Generate two different new relevant problems that are easier to solve than the example problem below.

Example problem:
Q: If a bag of marbles costs $20 and the price increases by 20% of the original price every two months, how much would a bag of marbles cost after 36 months?

New problem 1:
Q:

---

**Output1:**
If a bag of apples costs $5 and the price increases by 10% of the original price every month, how much would a bag of apples cost after 6 months?

New problem 2:
Q: A bookstore sells a book for $15, and the price decreases by 5% of the original price every week. How much would the book cost after 4 weeks?

---

**Input2:**
Generate one new problem that is analogous to the example problem below.

Example problem:
Q: If a bag of marbles costs $20 and the price increases by 20% of the original price every two months, how much would a bag of marbles cost after 36 months?

New problem 1:
Q:

---

**Output2:**
If a car's value depreciates by 10% of its original price every year, and the car was initially worth $30,000, what would be the value of the car after 5 years?

---

Table 22: Example 3: proxy queries generated with Mode 2 on GPT-3.5-turbo

**Input1:**
Generate two different new relevant problems that are easier to solve than the example problem below.

Example problem:
Q: Debra is monitoring a beehive to see how many bees come and go in a day. She sees 30 bees leave the hive in the first 6 hours, and then she sees 1/2 that many bees return in the next 6 hours. She sees two times as many bees as she saw first leave the hive fly from the hive and leave in the next 6 hours. Then every bee that left before that hadn't already returned returns to the hive in the next 6 hours. How many bees did Debra see return to the hive in the last 6 hours of the day?

New problem 1:
Q:

**Output1:**
A bakery sells 100 cupcakes in the morning, and then sells 1/4 of that number in the afternoon. Later in the evening, they sell twice as many cupcakes as they sold in the morning. How many cupcakes did the bakery sell in the evening?

New problem 2:
Q: A store initially has 50 shirts in stock. They sell 1/5 of the shirts in the morning, and then receive a new shipment of 30 shirts. In the afternoon, they sell twice as many shirts as they sold in the morning. How many shirts are left in the store at the end of the day?

**Input2:**
Generate one new problem that is analogous to the example problem below.

Example problem:
Q: Debra is monitoring a beehive to see how many bees come and go in a day. She sees 30 bees leave the hive in the first 6 hours, and then she sees 1/2 that many bees return in the next 6 hours. She sees two times as many bees as she saw first leave the hive fly from the hive and leave in the next 6 hours. Then every bee that left before that hadn't already returned returns to the hive in the next 6 hours. How many bees did Debra see return to the hive in the last 6 hours of the day?

New problem 1:
Q:

**Output2:**
Samantha is tracking the number of cars entering and exiting a parking lot. In the first hour, she counts 50 cars entering the parking lot. In the next hour, she observes 1/3 of that number leaving the parking lot. After that, she sees three times as many cars as she initially counted entering the parking lot and leaving in the following hour. Finally, all the cars that entered before and hadn't already left exit the parking lot in the last hour. How many cars did Samantha see exit the parking lot in the last hour of the day?

Table 23: Example 1: proxy queries generated with Mode 3 on GPT-3.5-turbo.

**Input:**
Generate three new analogous problems ranging from easy to hard for the example problem below, with the most challenging new problem being no more difficult than the example problem below.

Example problem:
Q: Lee used to be able to run the 400-meter hurdles two seconds faster than Gerald would run the 400-meter hurdles. But Gerald changed his diet, which improved his speed by 10%. If Lee runs the 400-meter hurdles in 38 seconds, how fast can Gerald, with his improved diet, run the 400-meter hurdles, in seconds?

New problem 1:
Q:

**Output:**
Sara used to be able to solve math problems 20% faster than John. But John started using a new study technique that improved his speed by 15%. If Sara can solve a math problem in 5 minutes, how fast can John, with his improved study technique, solve the same math problem, in minutes?

New problem 2:
Q: Alex used to be able to swim 100 meters faster than Mia by 10 seconds. But Mia started taking swimming lessons, which improved her speed by 20%. If Alex can swim 100 meters in 1 minute and 30 seconds, how fast can Mia, with her improved swimming skills, swim the same distance, in seconds?

New problem 3:
Q: Rachel used to be able to bake a cake in 45 minutes, which was 25% faster than Emily. But Emily learned a new baking technique that improved her speed by 30%. If Rachel can bake a cake in 45 minutes, how fast can Emily, with her improved baking technique, bake the same cake, in minutes?

Table 24: Example 2: proxy queries generated with Mode 3 on GPT-3.5-turbo.

**Input:**
Generate three new analogous problems ranging from easy to hard for the example problem below, with the most challenging new problem being no more difficult than the example problem below.

Example problem:
Q: If a bag of marbles costs $20 and the price increases by 20% of the original price every two months, how much would a bag of marbles cost after 36 months?

New problem 1:
Q:

**Output:**
If a bag of apples costs $5 and the price increases by 10% of the original price every month, how much would a bag of apples cost after 12 months?

New problem 2:
Q: If a car rental costs $50 per day and the price increases by $5 every week, how much would it cost to rent the car for 4 weeks?

New problem 3:
Q: If a concert ticket costs $100 and the price increases by 15% of the original price every three months, how much would a concert ticket cost after 24 months?

Table 25: Example 3: proxy queries generated with Mode 3 on GPT-3.5-turbo.

**Input:**
Generate three new analogous problems ranging from easy to hard for the example problem below, with the most challenging new problem being no more difficult than the example problem below.

Example problem:
Q: Debra is monitoring a beehive to see how many bees come and go in a day. She sees 30 bees leave the hive in the first 6 hours, and then she sees 1/2 that many bees return in the next 6 hours. She sees two times as many bees as she saw first leave the hive fly from the hive and leave in the next 6 hours. Then every bee that left before that hadn't already returned returns to the hive in the next 6 hours. How many bees did Debra see return to the hive in the last 6 hours of the day?

New problem 1:
Q:

**Output:**
Debra is monitoring a bird feeder to see how many birds come and go in a day. She sees 20 birds arrive at the feeder in the first 4 hours, and then she sees 1/3 that many birds leave in the next 4 hours. She sees three times as many birds as she saw first arrive at the feeder and leave in the next 4 hours. Then every bird that arrived before that hadn't already left leaves the feeder in the next 4 hours. How many birds did Debra see leave the feeder in the last 4 hours of the day?

New problem 2:
Q: Debra is monitoring a fish tank to see how many fish come and go in a day. She sees 50 fish swim out of the tank in the first 10 hours, and then she sees 1/4 that many fish return in the next 10 hours. She sees four times as many fish as she saw first swim out of the tank and leave in the next 10 hours. Then every fish that swam out before that hadn't already returned swims back into the tank in the next 10 hours. How many fish did Debra see return to the tank in the last 10 hours of the day?

New problem 3:
Q: Debra is monitoring a train station to see how many passengers come and go in a day. She sees 100 passengers board the train in the first 2 hours, and then she sees 1/5 that many passengers disembark in the next 2 hours. She sees five times as many passengers as she saw first board the train and disembark in the next 2 hours. Then every passenger who boarded before that hadn't already disembarked gets off the train in the next 2 hours. How many passengers did Debra see disembark from the train in the last 2 hours of the day?

Table 26: Example 1: proxy queries generated with Mode 1 on Llama3-70B-Instruct.

**Input:**
Generate two different new relevant problems that are easier to solve than the example problem below. And then generate one new problem that is analogous to the example problem below.

Example problem:
Q: Lee used to be able to run the 400-meter hurdles two seconds faster than Gerald would run the 400-meter hurdles. But Gerald changed his diet, which improved his speed by 10%. If Lee runs the 400-meter hurdles in 38 seconds, how fast can Gerald, with his improved diet, run the 400-meter hurdles, in seconds?

New problem 1:
Q:

**Output:**
Sarah can type 10 words per minute faster than her sister Emily. If Sarah can type 60 words per minute, how fast can Emily type, in words per minute?

New problem 2:
Q: A book costs $5 more than a magazine. If the book costs $15, how much does the magazine cost, in dollars?

Analogous problem:
Q: Tom used to be able to swim the 200-meter freestyle one minute faster than Rachel would swim the 200-meter freestyle. But Rachel started training with a new coach, which improved her speed by 12%. If Tom swims the 200-meter freestyle in 2 minutes and 10 seconds, how fast can Rachel, with her improved training, swim the 200-meter freestyle, in seconds?

Table 27: Example 2: proxy queries generated with Mode 1 on Llama3-70B-Instruct.

**Input:**
Generate two different new relevant problems that are easier to solve than the example problem below. And then generate one new problem that is analogous to the example problem below.

Example problem:
Q: If a bag of marbles costs $20 and the price increases by 20% of the original price every two months, how much would a bag of marbles cost after 36 months?

New problem 1:
Q:

**Output:**
If a book costs $15 and the price increases by $2 every year, how much would the book cost after 5 years?

New problem 2 :
Q: If a bicycle costs $80 and the price decreases by 10% every 6 months, how much would the bicycle cost after 18 months?

Analogous problem:
Q: If a laptop costs $1,200 and the price increases by 15% of the original price every 3 months, how much would the laptop cost after 24 months?

Table 28: Example 3: proxy queries generated with Mode 1 on Llama3-70B-Instruct.

**Input:**
Generate two different new relevant problems that are easier to solve than the example problem below. And then generate one new problem that is analogous to the example problem below.

Example problem:
Q: Debra is monitoring a beehive to see how many bees come and go in a day. She sees 30 bees leave the hive in the first 6 hours, and then she sees 1/2 that many bees return in the next 6 hours. She sees two times as many bees as she saw first leave the hive fly from the hive and leave in the next 6 hours. Then every bee that left before that hadn't already returned returns to the hive in the next 6 hours. How many bees did Debra see return to the hive in the last 6 hours of the day?

New problem 1:
Q:

**Output:**
A bookshelf has 5 shelves, and each shelf can hold 8 books. If the bookshelf is currently empty, how many books can be placed on it in total?

New problem 2:
Q: A bakery sells 12 loaves of bread in the morning and 8 loaves in the afternoon. How many loaves of bread did the bakery sell in total?

Analogous problem:
Q: A parking lot has 20 cars parked in it at 8am. Between 8am and 12pm, 1/4 of the cars leave the parking lot. Between 12pm and 2pm, 3 times as many cars as left earlier arrive at the parking lot. Between 2pm and 4pm, all the cars that left earlier return to the parking lot. How many cars are parked in the parking lot at 4pm?

Table 29: Example 1: proxy queries generated with Mode 2 on Llama3-70B-Instruct.

---

**Input1:**
Generate two different new relevant problems that are easier to solve than the example problem below.

Example problem:
Q: Lee used to be able to run the 400-meter hurdles two seconds faster than Gerald would run the 400-meter hurdles. But Gerald changed his diet, which improved his speed by 10%. If Lee runs the 400-meter hurdles in 38 seconds, how fast can Gerald, with his improved diet, run the 400-meter hurdles, in seconds?

New problem 1:
Q:

---

**Output1:**
Tom can paint a room in 4 hours, and his brother Alex can paint the same room in 5 hours. If Tom helps Alex, how many hours will it take them to paint the room together?

New problem 2:
Q: A bakery sells a total of 250 loaves of bread per day. They sell a combination of whole wheat and white bread. If they sell 30 more loaves of whole wheat than white bread, and each type of bread costs $2, how much money does the bakery make in a day from selling whole wheat bread?

---

**Input2:**
Generate one new problem that is analogous to the example problem below.

Example problem:
Q: Lee used to be able to run the 400-meter hurdles two seconds faster than Gerald would run the 400-meter hurdles. But Gerald changed his diet, which improved his speed by 10%. If Lee runs the 400-meter hurdles in 38 seconds, how fast can Gerald, with his improved diet, run the 400-meter hurdles, in seconds?

New problem 1:
Q:

---

**Output2:**
Maria used to be able to swim the 200-meter freestyle three seconds faster than Sophia would swim the 200-meter freestyle. But Sophia started training with a new coach, which improved her speed by 12%. If Maria swims the 200-meter freestyle in 1 minute 45 seconds, how fast can Sophia, with her improved training, swim the 200-meter freestyle, in seconds?

---

Table 30: Example 2: proxy queries generated with Mode 2 on Llama3-70B-Instruct

**Input1:**
Generate two different new relevant problems that are easier to solve than the example problem below.

Example problem:
Q: If a bag of marbles costs $20 and the price increases by 20% of the original price every two months, how much would a bag of marbles cost after 36 months?

New problem 1:
Q:

**Output1:**
A book costs $15 and the price increases by $2 every year. How much would the book cost after 5 years?

New problem 2:
Q: A bicycle costs $80 and the price decreases by 10% every year. How much would the bicycle cost after 3 years?

**Input2:**
Generate one new problem that is analogous to the example problem below.

Example problem:
Q: If a bag of marbles costs $20 and the price increases by 20% of the original price every two months, how much would a bag of marbles cost after 36 months?

New problem 1:
Q:

**Output2:**
If a gym membership costs $80 per month and the price increases by 15% of the original price every quarter, how much would a gym membership cost per month after 2 years?

Table 31: Example 3: proxy queries generated with Mode 2 on Llama3-70B-Instruct

---

**Input1:**
Generate two different new relevant problems that are easier to solve than the example problem below.

Example problem:
Q: Debra is monitoring a beehive to see how many bees come and go in a day. She sees 30 bees leave the hive in the first 6 hours, and then she sees 1/2 that many bees return in the next 6 hours. She sees two times as many bees as she saw first leave the hive fly from the hive and leave in the next 6 hours. Then every bee that left before that hadn't already returned returns to the hive in the next 6 hours. How many bees did Debra see return to the hive in the last 6 hours of the day?

New problem 1:
Q:

---

**Output1:**
Tom has 15 boxes of crayons in his art room. He gives 3 boxes to his friend Emma. Then, he receives 2 more boxes as a gift. How many boxes of crayons does Tom have now?

New problem 2:
Q: A bookshelf has 8 books on it. If 2 more books are added, and then 1 book is removed, how many books are on the bookshelf now?

---

**Input2:**
Generate one new problem that is analogous to the example problem below.

Example problem:
Q: Debra is monitoring a beehive to see how many bees come and go in a day. She sees 30 bees leave the hive in the first 6 hours, and then she sees 1/2 that many bees return in the next 6 hours. She sees two times as many bees as she saw first leave the hive fly from the hive and leave in the next 6 hours. Then every bee that left before that hadn't already returned returns to the hive in the next 6 hours. How many bees did Debra see return to the hive in the last 6 hours of the day?

New problem 1:
Q:

---

**Output2:**
Tom is tracking the number of books borrowed and returned to a library over a week. He sees 40 books borrowed in the first 2 days, and then he sees 3/4 that many books returned in the next 2 days. He sees 5 more books than he saw borrowed initially get borrowed in the next 2 days. Then every book that was borrowed but hadn't been returned yet is returned to the library in the last day of the week. How many books did Tom see returned to the library on the last day of the week?

---

Table 32: Example 1: proxy queries generated with Mode 3 on Llama3-70B-Instruct.

**Input:**
Generate three new analogous problems ranging from easy to hard for the example problem below, with the most challenging new problem being no more difficult than the example problem below.

Example problem:
Q: Lee used to be able to run the 400-meter hurdles two seconds faster than Gerald would run the 400-meter hurdles. But Gerald changed his diet, which improved his speed by 10%. If Lee runs the 400-meter hurdles in 38 seconds, how fast can Gerald, with his improved diet, run the 400-meter hurdles, in seconds?

New problem 1:
Q:

**Output:**
Sarah used to be able to swim the 200-meter freestyle one second faster than Emily would swim the 200-meter freestyle. But Emily started practicing more, which improved her speed by 5%. If Sarah swims the 200-meter freestyle in 25 seconds, how fast can Emily, with her improved practice, swim the 200-meter freestyle, in seconds?

New problem 2:
Q: Alex used to be able to solve a Rubik's Cube 3 seconds faster than Ben would solve a Rubik's Cube. But Ben started using a new solving method, which improved his speed by 12%. If Alex solves a Rubik's Cube in 45 seconds, how fast can Ben, with his new method, solve a Rubik's Cube, in seconds?

New problem 3:
Q: Rachel used to be able to complete a 5K bike ride 2 minutes faster than Michael would complete a 5K bike ride. But Michael started training on hills, which improved his speed by 15%. If Rachel completes a 5K bike ride in 12 minutes, how fast can Michael, with his hill training, complete a 5K bike ride, in minutes?

Table 33: Example 2: proxy queries generated with Mode 3 on Llama3-70B-Instruct.

**Input:**
Generate three new analogous problems ranging from easy to hard for the example problem below, with the most challenging new problem being no more difficult than the example problem below.

Example problem:
Q: If a bag of marbles costs $20 and the price increases by 20% of the original price every two months, how much would a bag of marbles cost after 36 months?

New problem 1:
Q:

**Output:**
If a book costs $15 and the price increases by 15% of the original price every year, how much would the book cost after 4 years?

New problem 2:
Q: If a bike costs $80 and the price increases by 25% of the original price every quarter, how much would the bike cost after 18 months?

New problem 3:
Q: If a laptop costs $1200 and the price increases by 18% of the original price every 3 months, how much would the laptop cost after 42 months?

Table 34: Example 3: proxy queries generated with Mode 3 on Llama3-70B-Instruct.

**Input:**
Generate three new analogous problems ranging from easy to hard for the example problem below, with the most challenging new problem being no more difficult than the example problem below.

Example problem:
Q: Debra is monitoring a beehive to see how many bees come and go in a day. She sees 30 bees leave the hive in the first 6 hours, and then she sees 1/2 that many bees return in the next 6 hours. She sees two times as many bees as she saw first leave the hive fly from the hive and leave in the next 6 hours. Then every bee that left before that hadn't already returned returns to the hive in the next 6 hours. How many bees did Debra see return to the hive in the last 6 hours of the day?

New problem 1:
Q:

**Output:**
A bookshelf has 12 books on it. In the morning, 4 books are removed. In the afternoon, 2 books are added. In the evening, 8 books are added. How many books are on the bookshelf at the end of the day?

New problem 2:
Q: A farmer has 18 baskets of apples. He gives away 6 baskets to his neighbor. Then, he receives 3 baskets of apples from another farmer. Later, he sells 9 baskets of apples at the market. How many baskets of apples does the farmer have left?

New problem 3:
Q: A water tank can hold 24 gallons of water. In the morning, 8 gallons of water are pumped into the tank. In the afternoon, 1/3 of the water in the tank is drained. In the evening, 12 gallons of water are pumped into the tank. How many gallons of water are in the tank at the end of the day?

Table 35: Example 1: proxy queries generated with Mode 1 on Qwen1.5-72B-Chat.

---

**Input:**
Generate two different new relevant problems that are easier to solve than the example problem below. And then generate one new problem that is analogous to the example problem below.

Example problem:
Q: Lee used to be able to run the 400-meter hurdles two seconds faster than Gerald would run the 400-meter hurdles. But Gerald changed his diet, which improved his speed by 10%. If Lee runs the 400-meter hurdles in 38 seconds, how fast can Gerald, with his improved diet, run the 400-meter hurdles, in seconds?

New problem 1:
Q:

---

**Output:**
Sarah can paint a room in 6 hours, while David can do it in 4 hours. If they work together, how long will it take them to paint the same room?

New problem 2:
Q: A car travels at an average speed of 60 km/h on a highway. If the driver increases the speed by 20%, how many kilometers further can they travel in the same amount of time as before?

Analogous problem:
Q: Olivia can complete a 10-kilometer bike race 5 minutes faster than Jake. Recently, Jake started training with a coach and improved his time by 15%. If Olivia finishes the race in 35 minutes, how long will it take Jake, with his improved performance, to complete the 10-kilometer race?

---

Table 36: Example 2: proxy queries generated with Mode 1 on Qwen1.5-72B-Chat.

---

**Input:**
Generate two different new relevant problems that are easier to solve than the example problem below. And then generate one new problem that is analogous to the example problem below.

Example problem:
Q: If a bag of marbles costs $20 and the price increases by 20% of the original price every two months, how much would a bag of marbles cost after 36 months?

New problem 1:
Q:

---

**Output:**
A book is initially priced at $15 and its price decreases by 10% of its current price every three weeks. What will be the price of the book after 8 weeks?

New problem 2 :
Q: A box of cookies retails for $12, and there's a 15% discount on the box if purchased within a week. How much would a customer save if they buy the box on the sixth day?

Analogous problem:
Q: A bicycle is priced at $400 and the store offers a 15% discount on the original price every four months. What would be the final price of the bicycle after 32 months?

---

Table 37: Example 3: proxy queries generated with Mode 1 on Qwen1.5-72B-Chat.

**Input:**
Generate two different new relevant problems that are easier to solve than the example problem below. And then generate one new problem that is analogous to the example problem below.

Example problem:
Q: Debra is monitoring a beehive to see how many bees come and go in a day. She sees 30 bees leave the hive in the first 6 hours, and then she sees 1/2 that many bees return in the next 6 hours. She sees two times as many bees as she saw first leave the hive fly from the hive and leave in the next 6 hours. Then every bee that left before that hadn't already returned returns to the hive in the next 6 hours. How many bees did Debra see return to the hive in the last 6 hours of the day?

New problem 1:
Q:

**Output:**
A library has received 50 new books in the morning. By noon, 30 books have been checked out by patrons. In the afternoon, half of those books are returned. Later in the day, another batch of 20 books is added to the shelves. How many books are there in total at the end of the day?

New problem 2:
Q: A farmer plants 40 tomato seedlings in the morning. By noon, 25% of them have sprouted. In the afternoon, he adds another 15 seedlings. At the end of the day, all the sprouted seedlings grow an additional leaf. How many tomato seedlings have at least one leaf by the end of the day?

Analogous problem:
Q: A school bus picks up students in the morning. On the first route, it collects 30 students. In the afternoon, 15 students disembark at their homes, which is half the number picked up initially. The bus then picks up twice the number of students it dropped off on its second route. Finally, all the remaining students who were picked up earlier return home on the last route. How many students did the bus drop off at their homes during the last route of the day?

Table 38: Example 1: proxy queries generated with Mode 2 on Qwen1.5-72B-Chat.

**Input1:**
Generate two different new relevant problems that are easier to solve than the example problem below.

Example problem:
Q: Lee used to be able to run the 400-meter hurdles two seconds faster than Gerald would run the 400-meter hurdles. But Gerald changed his diet, which improved his speed by 10%. If Lee runs the 400-meter hurdles in 38 seconds, how fast can Gerald, with his improved diet, run the 400-meter hurdles, in seconds?

New problem 1:
Q:

**Output1:**
A library has twice as many books as a smaller branch. If the smaller branch received a donation of 50 books, it now has a total of 200 books. How many books are there in the main library?

New problem 2:
Q: A bakery sells cupcakes in boxes of 6 and cookies in boxes of 12. If a customer buys an equal number of cupcake boxes and cookie boxes, they get a 10% discount on the total box cost. If one cupcake box costs $10 and one cookie box costs $15, how much does the customer pay after the discount if they buy enough boxes for 72 cupcakes and 72 cookies?

**Input2:**
Generate one new problem that is analogous to the example problem below.

Example problem:
Q: Lee used to be able to run the 400-meter hurdles two seconds faster than Gerald would run the 400-meter hurdles. But Gerald changed his diet, which improved his speed by 10%. If Lee runs the 400-meter hurdles in 38 seconds, how fast can Gerald, with his improved diet, run the 400-meter hurdles, in seconds?

New problem 1:
Q:

**Output2:**
Sarah typically completes a 5-kilometer jog 2 minutes quicker than Mike. Recently, Mike started using a fitness app that increased his speed by 15%. If Sarah finishes the 5-kilometer jog in 24 minutes, how long, in minutes, does it take for Mike to complete the jog with his improved pace?

Table 39: Example 2: proxy queries generated with Mode 2 on Qwen1.5-72B-Chat

| |
|---|
| **Input1:** |
| Generate two different new relevant problems that are easier to solve than the example problem below. |
| |
| Example problem: |
| Q: If a bag of marbles costs $20 and the price increases by 20% of the original price every two months, how much would a bag of marbles cost after 36 months? |
| |
| New problem 1: |
| Q: |
| **Output1:** |
| A book is initially priced at $15. If the bookstore offers a discount of 10% on the book every six months, what would be the price of the book after one year? |
| |
| New problem 2: |
| Q: A monthly subscription service starts at $10 per month. If the company raises the price by $1 every four months, what will the subscription cost after 12 months? |
| **Input2:** |
| Generate one new problem that is analogous to the example problem below. |
| |
| Example problem: |
| Q: If a bag of marbles costs $20 and the price increases by 20% of the original price every two months, how much would a bag of marbles cost after 36 months? |
| |
| New problem 1: |
| Q: |
| **Output2:** |
| A book costs $15 and the price increases by 15% of its current value every six months. What would be the cost of the book after 24 months? |

Table 40: Example 3: proxy queries generated with Mode 2 on Qwen1.5-72B-Chat

**Input1:**
Generate two different new relevant problems that are easier to solve than the example problem below.

Example problem:
Q: Debra is monitoring a beehive to see how many bees come and go in a day. She sees 30 bees leave the hive in the first 6 hours, and then she sees 1/2 that many bees return in the next 6 hours. She sees two times as many bees as she saw first leave the hive fly from the hive and leave in the next 6 hours. Then every bee that left before that hadn't already returned returns to the hive in the next 6 hours. How many bees did Debra see return to the hive in the last 6 hours of the day?

New problem 1:
Q:

**Output1:**
A library has 500 books on its shelves. In the morning, 150 books are borrowed by patrons. Throughout the day, 75 books are returned. In the afternoon, another 100 books are borrowed. How many books are not yet returned to the library at the end of the day?

New problem 2:
Q: A bakery made 400 cupcakes for the day. In the morning, they sold 150 cupcakes. During lunchtime, they sold half of what they sold in the morning. In the afternoon, they sold 120 more cupcakes. How many cupcakes did the bakery have left unsold at the end of the day?

**Input2:**
Generate one new problem that is analogous to the example problem below.

Example problem:
Q: Debra is monitoring a beehive to see how many bees come and go in a day. She sees 30 bees leave the hive in the first 6 hours, and then she sees 1/2 that many bees return in the next 6 hours. She sees two times as many bees as she saw first leave the hive fly from the hive and leave in the next 6 hours. Then every bee that left before that hadn't already returned returns to the hive in the next 6 hours. How many bees did Debra see return to the hive in the last 6 hours of the day?

New problem 1:
Q:

**Output2:**
Henry is studying the migration patterns of butterflies in his garden. He notices that 50 butterflies take off in the morning and fly away. In the afternoon, half of those butterflies come back. Later, he observes twice the number of butterflies that initially flew away fluttering out of the garden. Finally, all the butterflies that had left earlier return before sunset. How many butterflies did Henry see returning to the garden during the last part of the day?

Table 41: Example 1: proxy queries generated with Mode 3 on Qwen1.5-72B-Chat.

**Input:**
Generate three new analogous problems ranging from easy to hard for the example problem below, with the most challenging new problem being no more difficult than the example problem below.

Example problem:
Q: Lee used to be able to run the 400-meter hurdles two seconds faster than Gerald would run the 400-meter hurdles. But Gerald changed his diet, which improved his speed by 10%. If Lee runs the 400-meter hurdles in 38 seconds, how fast can Gerald, with his improved diet, run the 400-meter hurdles, in seconds?

New problem 1:
Q:

**Output:**
Sarah can paint a room in 6 hours while David takes 2 hours more to complete painting the same room. Recently, David started using a new painting technique that increased his efficiency by 25%. If Sarah paints at a constant rate, how long would it take for David, with his improved technique, to paint the room?

New problem 2:
Q: Jake can type 500 words in 30 minutes, while Mia types 20% fewer words in the same time. Mia recently started using a typing software that increased her typing speed by 30%. If Jake's typing speed remains constant, how many words can Mia type in 30 minutes after improving her typing speed?

New problem 3:
Q: A delivery van can cover a distance of 200 km in 2 hours. A car, on the other hand, takes 30 minutes longer to cover the same distance. The car's driver recently upgraded its engine, resulting in a 15% improvement in its average speed. If the delivery van maintains its speed, how long would it take for the upgraded car to cover 200 km?

Table 42: Example 2: proxy queries generated with Mode 3 on Qwen1.5-72B-Chat.

**Input:**
Generate three new analogous problems ranging from easy to hard for the example problem below, with the most challenging new problem being no more difficult than the example problem below.

Example problem:
Q: If a bag of marbles costs $20 and the price increases by 20% of the original price every two months, how much would a bag of marbles cost after 36 months?

New problem 1:
Q:

**Output:**
A book is initially priced at $15 and the price increases by 10% of its original price every six months. What will be the price of the book after 4 years?

New problem 2:
Q: A box of chocolates costs $30 and undergoes a price increase of 15% of its original price every three months. How much would it cost after 30 months?

New problem 3:
Q: A smartphone is sold for $800 and its price decreases by 10% of its original value every year due to depreciation. What will be the price of the smartphone after 5 years?

Table 43: Example 3: proxy queries generated with Mode 3 on Qwen1.5-72B-Chat.

**Input:**
Generate three new analogous problems ranging from easy to hard for the example problem below, with the most challenging new problem being no more difficult than the example problem below.

Example problem:
Q: Debra is monitoring a beehive to see how many bees come and go in a day. She sees 30 bees leave the hive in the first 6 hours, and then she sees 1/2 that many bees return in the next 6 hours. She sees two times as many bees as she saw first leave the hive fly from the hive and leave in the next 6 hours. Then every bee that left before that hadn't already returned returns to the hive in the next 6 hours. How many bees did Debra see return to the hive in the last 6 hours of the day?

New problem 1:
Q:

**Output:**
A birdwatcher observes a flock of birds at a park. In the morning, he counts 25 birds flying away. Later, he sees half of that number returning. In the afternoon, he notices another group of birds, which is twice the size of the initial group that flew away, leaving the park. Finally, all the birds that had left earlier come back. How many birds did the birdwatcher see return in the evening?

New problem 2:
Q: At a farmer's market, a vendor sells apples and oranges. In the morning, they sell 40 apples. By noon, they sell half that amount in oranges. In the afternoon, they sell twice as many bananas as the total number of apples sold in the morning. Towards the end of the day, all the customers who bought fruits earlier return, and each buys an equal number of apples and oranges. How many apples did the vendor sell to returning customers?

New problem 3:
Q: A library has a set of books for loan. In the morning, 50 books are borrowed by patrons. By afternoon, half of those books are returned. In the evening, another batch of books is borrowed, this time twice the number of the initial borrowings. Finally, all the remaining books that were borrowed earlier are returned. How many books did the librarian check in during the last borrowing session of the day?

Table 44: Example 1: proxy queries generated with Mode 1 on Qwen1.5-14B-Chat.

**Input:**
Generate two different new relevant problems that are easier to solve than the example problem below. And then generate one new problem that is analogous to the example problem below.

Example problem:
Q: Lee used to be able to run the 400-meter hurdles two seconds faster than Gerald would run the 400-meter hurdles. But Gerald changed his diet, which improved his speed by 10%. If Lee runs the 400-meter hurdles in 38 seconds, how fast can Gerald, with his improved diet, run the 400-meter hurdles, in seconds?

New problem 1:
Q:

**Output:**
Alice can paint a room in 6 hours, while Bob can do it in half the time. If they work together, how long will it take them to paint the room?

New problem 2:
Q: A library has 500 books. If it adds 20% more books every month, how many books will there be after 3 months?

Analogous problem:
Q: Sarah can type 70 words per minute, while her friend Emily types 20% faster. If Sarah takes 15 minutes to complete a task, how long will it take Emily, with her increased typing speed, to finish the same task?

Table 45: Example 2: proxy queries generated with Mode 1 on Qwen1.5-14B-Chat.

**Input:**
Generate two different new relevant problems that are easier to solve than the example problem below. And then generate one new problem that is analogous to the example problem below.

Example problem:
Q: If a bag of marbles costs $20 and the price increases by 20% of the original price every two months, how much would a bag of marbles cost after 36 months?

New problem 1:
Q:

**Output:**
A jar contains 40 candies, and each day, 5 more candies are added. How many candies will be in the jar after 7 days?

New problem 2 :
Q: A garden has a rectangular shape with a length of 15 meters and a width of 8 meters. If the gardener doubles the length every year, what will be the new area after 3 years?

Analogous problem:
Q: A person saves $50 per month in a bank account with a 3% annual interest rate, compounded monthly. How much money will they have in the account after 2 years?

Table 46: Example 3: proxy queries generated with Mode 1 on Qwen1.5-14B-Chat.

**Input:**
Generate two different new relevant problems that are easier to solve than the example problem below. And then generate one new problem that is analogous to the example problem below.

Example problem:
Q: Debra is monitoring a beehive to see how many bees come and go in a day. She sees 30 bees leave the hive in the first 6 hours, and then she sees 1/2 that many bees return in the next 6 hours. She sees two times as many bees as she saw first leave the hive fly from the hive and leave in the next 6 hours. Then every bee that left before that hadn't already returned returns to the hive in the next 6 hours. How many bees did Debra see return to the hive in the last 6 hours of the day?

New problem 1:
Q:

**Output:**
A family goes on a picnic, and they bring 15 apples. They eat half of them at lunchtime. Later, they find 4 more apples under a tree. How many apples do they have left after eating another quarter of the total?

New problem 2:
Q: A school library has 200 books. On Monday, students borrow 40% of the books. On Tuesday, they return 30% of what they borrowed. How many books are still available on Tuesday?

Analogous problem:
Q: A gardener starts with 75 plants in her garden. In the morning, she waters half of them, and during the afternoon, she adds 15 more plants. Later, some plants wilt and she removes 25% of the ones she watered in the morning. How many plants does she have at the end of the day?

Table 47: Example 1: proxy queries generated with Mode 2 on Qwen1.5-14B-Chat.

**Input1:**
Generate two different new relevant problems that are easier to solve than the example problem below.

Example problem:
Q: Lee used to be able to run the 400-meter hurdles two seconds faster than Gerald would run the 400-meter hurdles. But Gerald changed his diet, which improved his speed by 10%. If Lee runs the 400-meter hurdles in 38 seconds, how fast can Gerald, with his improved diet, run the 400-meter hurdles, in seconds?

New problem 1:
Q:

**Output1:**
Alicia can paint a small room in 4 hours, while her friend Ben can do the same job in 6 hours. How long will it take for them to paint the room together if they work simultaneously?

New problem 2:
Q: If it takes Jane 30 minutes to walk to school and back, and her friend Sarah rides her bike at twice the speed, how long does it take Sarah to make the round trip to school?

**Input2:**
Generate one new problem that is analogous to the example problem below.

Example problem:
Q: Lee used to be able to run the 400-meter hurdles two seconds faster than Gerald would run the 400-meter hurdles. But Gerald changed his diet, which improved his speed by 10%. If Lee runs the 400-meter hurdles in 38 seconds, how fast can Gerald, with his improved diet, run the 400-meter hurdles, in seconds?

New problem 1:
Q:

**Output2:**
Maria was initially able to paint a mural twice as large as Sarah's in the same amount of time. After joining an art class, Sarah increased her painting speed by 30%. If Maria now completes her mural in 6 hours, how long would it take Sarah, with her improved skills, to paint her mural if it originally took her 8 hours?

Table 48: Example 2: proxy queries generated with Mode 2 on Qwen1.5-14B-Chat

**Input1:**
Generate two different new relevant problems that are easier to solve than the example problem below.

Example problem:
Q: If a bag of marbles costs $20 and the price increases by 20% of the original price every two months, how much would a bag of marbles cost after 36 months?

New problem 1:
Q:

**Output1:**
A box contains 40 apples. If a person eats 3 apples each day, how many days will it take to finish the box?

New problem 2:
Q: A garden has a rectangular shape with a length of 15 meters and a width of 8 meters. What is the perimeter of the garden?

**Input2:**
Generate one new problem that is analogous to the example problem below.

Example problem:
Q: If a bag of marbles costs $20 and the price increases by 20% of the original price every two months, how much would a bag of marbles cost after 36 months?

New problem 1:
Q:

**Output2:**
If a book's initial price is $15 and its price decreases by 15% of the current price every six months, how much would the book cost after 24 months?

Table 49: Example 3: proxy queries generated with Mode 2 on Qwen1.5-14B-Chat

**Input1:**
Generate two different new relevant problems that are easier to solve than the example problem below.

Example problem:
Q: Debra is monitoring a beehive to see how many bees come and go in a day. She sees 30 bees leave the hive in the first 6 hours, and then she sees 1/2 that many bees return in the next 6 hours. She sees two times as many bees as she saw first leave the hive fly from the hive and leave in the next 6 hours. Then every bee that left before that hadn't already returned returns to the hive in the next 6 hours. How many bees did Debra see return to the hive in the last 6 hours of the day?

New problem 1:
Q:

**Output1:**
A family has 4 apples and they eat 2 apples each day. After how many days will there be no more apples left?

New problem 2:
Q: Sarah plants 5 flower seeds in her garden. If 3 seeds sprout, and she adds 2 more seeds that also sprout, how many seeds did she successfully grow?

**Input2:**
Generate one new problem that is analogous to the example problem below.

Example problem:
Q: Debra is monitoring a beehive to see how many bees come and go in a day. She sees 30 bees leave the hive in the first 6 hours, and then she sees 1/2 that many bees return in the next 6 hours. She sees two times as many bees as she saw first leave the hive fly from the hive and leave in the next 6 hours. Then every bee that left before that hadn't already returned returns to the hive in the next 6 hours. How many bees did Debra see return to the hive in the last 6 hours of the day?

New problem 1:
Q:

**Output2:**
John is observing a school garden where plants grow and wither throughout the week. On Monday, he notices 40 plants flowering. By Wednesday, half of those flowers have wilted, but on Thursday, a third of the remaining flowers bloom again. On Friday, all the plants that had wilted earlier regain their vitality. If he initially counted 80 plants in total, how many plants did John see bloom on Friday?

Table 50: Example 1: proxy queries generated with Mode 3 on Qwen1.5-14B-Chat.

**Input:**
Generate three new analogous problems ranging from easy to hard for the example problem below, with the most challenging new problem being no more difficult than the example problem below.

Example problem:
Q: Lee used to be able to run the 400-meter hurdles two seconds faster than Gerald would run the 400-meter hurdles. But Gerald changed his diet, which improved his speed by 10%. If Lee runs the 400-meter hurdles in 38 seconds, how fast can Gerald, with his improved diet, run the 400-meter hurdles, in seconds?

New problem 1:
Q:

**Output:**
Jane can paint a small room in half the time it takes her brother Tom to paint a similar room. If Jane completes a small room in 4 hours, how long will it take Tom, with his improved painting efficiency (25% faster), to paint the same room?

New problem 2:
Q: Alex used to solve a math puzzle in 15 minutes, which was 20% less time than it took his friend Ben. If Alex now solves the puzzle in 12 minutes due to practice, how much time does Ben, with his unchanged solving speed, need to complete the puzzle?

New problem 3:
Q: In a relay race, Team A's runner A completes a lap 15% faster than Team B's runner B. If Runner A finishes the lap in 45 seconds, how long does Runner B take to finish the same lap, considering his speed remains constant?

Table 51: Example 2: proxy queries generated with Mode 3 on Qwen1.5-14B-Chat.

**Input:**
Generate three new analogous problems ranging from easy to hard for the example problem below, with the most challenging new problem being no more difficult than the example problem below.

Example problem:
Q: If a bag of marbles costs $20 and the price increases by 20% of the original price every two months, how much would a bag of marbles cost after 36 months?

New problem 1:
Q:

**Output:**
If a pencil costs $1 and its price doubles every 6 months, how much will a pencil cost after 18 months?

New problem 2:
Q: A book's initial price is $15, and it increases by 15% each year. What will be the book's price after 5 years?

New problem 3:
Q: A gym membership starts at $49 per month, with a 10% monthly increase in fees. How much would you pay for a 2-year membership?

Table 52: Example 3: proxy queries generated with Mode 3 on Qwen1.5-14B-Chat.

**Input:**
Generate three new analogous problems ranging from easy to hard for the example problem below, with the most challenging new problem being no more difficult than the example problem below.

Example problem:
Q: Debra is monitoring a beehive to see how many bees come and go in a day. She sees 30 bees leave the hive in the first 6 hours, and then she sees 1/2 that many bees return in the next 6 hours. She sees two times as many bees as she saw first leave the hive fly from the hive and leave in the next 6 hours. Then every bee that left before that hadn't already returned returns to the hive in the next 6 hours. How many bees did Debra see return to the hive in the last 6 hours of the day?

New problem 1:
Q:

**Output:**
A gardener observes butterflies in a garden. In the morning, 15 butterflies leave, and half that number return by noon. Later, 75% of the morning leavers return in the afternoon. How many butterflies returned in the afternoon?

New problem 2:
Q: At a library, 40 books are checked out in the morning. In the afternoon, half of the morning checkouts return, and an additional 10 books are returned by students who forgot earlier. How many books were returned in the afternoon?

New problem 3:
Q: A bakery prepares 80 loaves of bread in the morning. In the afternoon, 60% of the morning production is sold, and then 75% of the unsold loaves are returned to the bakery. How many loaves were returned in the afternoon if there were still 24 loaves left unsold?

Table 53: Example 1: Generation of easy- and hard-proxy queries by SPG and APG in LBS3 - (GSM8K, $n_1 = 1$)

**Input:**
Generate one new relevant problem that is easier to solve than the example problem below. And then generate two new problems that are analogous to the example problem below.

Example problem:
Q: Natalia sold clips to 48 of her friends in April, and then she sold half as many clips in May. How many clips did Natalia sell altogether in April and May?

New problem 1:
Q:

**Output:**
John read 20 pages of a book in the morning, and 30 pages in the afternoon. How many pages did John read in total?

New problem 2:
Q: Marcus baked cookies for 36 of his classmates in March, and then he baked twice as many cookies in April. How many cookies did Marcus bake altogether in March and April?

Analogous problem:
Q: Emily read books to 24 children in January, and then she read to three times as many children in February. How many children did Emily read books to altogether in January and February?

---

Table 54: Example 1: Using RAG to solve a easy-proxy query in LBS3-(GSM8K, $n_1 = 1$)

**Input:**
Solve the following example problem and present the final answer after "The answer is".

Example problem:
Q: John read 20 pages of a book in the morning, and 30 pages in the afternoon. How many pages did John read in total?
A: Let's think step by step.

**Output:**
First, we know John read 20 pages in the morning.
Next, he read an additional 30 pages in the afternoon.
To find the total number of pages John read, we need to add the pages from the morning and afternoon together:
20 pages + 30 pages = 50 pages

The answer is 50 pages.

---

Table 61: Example 2: Using RAG to solve the first hard-proxy query in LBS3-(SQA, $n_1 = 2$)

---

**Input:**
Given examples:
Q: Does a cat meow more often than a dog barks?
A: Let's think step by step.
To solve this question, we need to consider the typical behaviors of cats and dogs, particularly focusing on their vocalization habits.
1. Understanding Cat Behavior: Cats meow for various reasons, such as to greet people, to seek attention, to ask for food, or to indicate distress. However, adult cats typically do not meow at each other, but rather reserve this sound for communicating with humans. The frequency of meowing can vary widely depending on the individual cat's personality, age, and circumstances.
2. Understanding Dog Behavior: Dogs bark as a form of communication with both humans and other dogs. The reasons for barking can include alerting to danger, expressing excitement, seeking attention, responding to other dogs, or as a response to boredom or anxiety. Like cats, the frequency of barking varies widely among individual dogs, influenced by breed, training, environment, and temperament.
3. Comparing the Two: Generally, dogs are considered to be more vocal than cats. Dogs use barking as a primary means of communication, and it is a behavior reinforced by both natural instincts and human training. Cats, while they do meow to communicate with humans, often use subtler forms of communication (like body language and scent marking) when interacting with other cats or when expressing themselves in their environment.
4. Conclusion: While there are exceptions depending on individual behavior and circumstances, generally speaking, dogs bark more frequently than cats meow. Dogs are often encouraged to bark for training and protective purposes, whereas cats typically meow less and often only in specific contexts (like interacting with humans).
The answer is: No, generally a cat does not meow more often than a dog barks.

Q: Can a rabbit run faster than a turtle?
A: Let's think step by step. To determine whether a rabbit can run faster than a turtle, we should consider the general known speeds of both animals.
1. Speed of a Rabbit: Rabbits are known for their ability to run quite fast to escape predators. The average speed of a wild rabbit is about 25-30 miles per hour (mph). Some rabbits can reach these top speeds so that they can quickly escape from threats.
2. Speed of a Turtle: Turtles, on the other hand, are known for their slow movement. The average walking speed of a typical turtle is approximately 0.2-0.5 miles per hour. Even when they feel threatened, turtles do not speed up significantly but rather retreat into their shells for protection.
3. Comparison: Comparing the speeds, 25-30 mph for rabbits vastly exceeds the 0.2-0.5 mph speed of a turtle.
Therefore, by analyzing the average speeds of both animals, it is clear that a rabbit can indeed run much faster than a turtle.
The answer is: Yes, a rabbit can run faster than a turtle.

Solve the following problem based on given examples above and present the final answer after "The answer is".

Q: Would a cat react to a whistle before a dolphin?
A: Let's think step by step.

**Output:**
To determine whether a cat would react to a whistle before a dolphin, we need to consider the hearing capabilities and response behaviors of both animals.

1. Understanding Cat Hearing and Response: Cats have excellent hearing and can detect a wide range of frequencies, from about 48 Hz to 85 kHz. They are particularly sensitive to high-pitched sounds, which makes them good at hearing and responding to whistles that are within their hearing range. Cats use their hearing to hunt and navigate their environment, but their response to a whistle can vary. Some cats may be trained to respond to specific sounds, while others might ignore them if they do not associate the sound with anything beneficial or interesting.

2. Understanding Dolphin Hearing and Response: Dolphins are renowned for their exceptional hearing abilities, crucial for echolocation. They can hear frequencies from about 75 Hz to 150 kHz, which is even broader than cats. Dolphins are highly responsive to auditory stimuli, including whistles, especially since trainers often use whistles in marine parks and research settings to communicate with them. Dolphins can quickly learn to associate whistles with specific behaviors or rewards.

3. Comparing Response Times: While both animals have excellent hearing, the context in which they respond to sounds like whistles is crucial. Domestic cats might not always see a need to respond immediately to a whistle unless trained to do so. Dolphins, however, are often trained to respond to whistles and do so very quickly as part of their learned behaviors in interactive environments.

4. Conclusion: Considering the typical training and environments of both animals, a dolphin is likely to react to a whistle more promptly than a cat. This is due to the dolphin's training to associate whistles with immediate actions or rewards, whereas a cat's reaction is less predictable and dependent on individual training and personality.

The answer is: No, a cat would not react to a whistle before a dolphin.

Table 62: Example 2: Using RAG to solve the target query in LBS3-(SQA, $n_1 = 2$)

**Input:**
Given examples:
Q: Does a cat meow more often than a dog barks?
A: Let's think step by step.
To solve this question, we need to consider the typical behaviors of cats and dogs, particularly focusing on their vocalization habits.
1. Understanding Cat Behavior: Cats meow for various reasons, such as to greet people, to seek attention, to ask for food, or to indicate distress. However, adult cats typically do not meow at each other, but rather reserve this sound for communicating with humans. The frequency of meowing can vary widely depending on the individual cat's personality, age, and circumstances.
2. Understanding Dog Behavior: Dogs bark as a form of communication with both humans and other dogs. The reasons for barking can include alerting to danger, expressing excitement, seeking attention, responding to other dogs, or as a response to boredom or anxiety. Like cats, the frequency of barking varies widely among individual dogs, influenced by breed, training, environment, and temperament.
3. Comparing the Two: Generally, dogs are considered to be more vocal than cats. Dogs use barking as a primary means of communication, and it is a behavior reinforced by both natural instincts and human training. Cats, while they do meow to communicate with humans, often use subtler forms of communication (like body language and scent marking) when interacting with other cats or when expressing themselves in their environment.
4. Conclusion: While there are exceptions depending on individual behavior and circumstances, generally speaking, dogs bark more frequently than cats meow. Dogs are often encouraged to bark for training and protective purposes, whereas cats typically meow less and often only in specific contexts (like interacting with humans).
The answer is: No, generally a cat does not meow more often than a dog barks.

Q: Can a rabbit run faster than a turtle?
A: Let's think step by step. To determine whether a rabbit can run faster than a turtle, we should consider the general known speeds of both animals.
1. Speed of a Rabbit: Rabbits are known for their ability to run quite fast to escape predators. The average speed of a wild rabbit is about 25-30 miles per hour (mph). Some rabbits can reach these top speeds so that they can quickly escape from threats.
2. Speed of a Turtle: Turtles, on the other hand, are known for their slow movement. The average walking speed of a typical turtle is approximately 0.2-0.5 miles per hour. Even when they feel threatened, turtles do not speed up significantly but rather retreat into their shells for protection.
3. Comparison: Comparing the speeds, 25-30 mph for rabbits vastly exceeds the 0.2-0.5 mph speed of a turtle.
Therefore, by analyzing the average speeds of both animals, it is clear that a rabbit can indeed run much faster than a turtle.
The answer is: Yes, a rabbit can run faster than a turtle.

Q: Would a cat react to a whistle before a dolphin?
A: Let's think step by step. To determine whether a cat would react to a whistle before a dolphin, we need to consider the hearing capabilities and response behaviors of both animals.
1. Understanding Cat Hearing and Response: Cats have excellent hearing and can detect a wide range of frequencies, from about 48 Hz to 85 kHz. They are particularly sensitive to high-pitched sounds, which makes them good at hearing and responding to whistles that are within their hearing range. Cats use their hearing to hunt and navigate their environment, but their response to a whistle can vary. Some cats may be trained to respond to specific sounds, while others might ignore them if they do not associate the sound with anything beneficial or interesting.
2. Understanding Dolphin Hearing and Response: Dolphins are renowned for their exceptional hearing abilities, crucial for echolocation. They can hear frequencies from about 75 Hz to 150 kHz, which is even broader than cats. Dolphins are highly responsive to auditory stimuli, including whistles, especially since trainers often use whistles in marine parks and research settings to communicate with them. Dolphins can quickly learn to associate whistles with specific behaviors or rewards.
3. Comparing Response Times: While both animals have excellent hearing, the context in which they respond to sounds like whistles is crucial. Domestic cats might not always see a need to respond immediately to a whistle unless trained to do so. Dolphins, however, are often trained to respond to whistles and do so very quickly as part of their learned behaviors in interactive environments.

4. Conclusion: Considering the typical training and environments of both animals, a dolphin is likely to react to a whistle more promptly than a cat. This is due to the dolphin's training to associate whistles with immediate actions or rewards, whereas a cat's reaction is less predictable and dependent on individual training and personality.
The answer is: No, a cat would not react to a whistle before a dolphin.

Solve the following problem based on given examples above and present the final answer after "The answer is".
Q: Would a dog respond to bell before Grey seal?
A: Let's think step by step.

**Output:**
To determine whether a dog would respond to a bell before a Grey seal, we need to consider the hearing capabilities, typical behavioral training, and reaction times of both animals.
1. Understanding Dog Hearing and Response: Dogs have a well-developed sense of hearing and can detect sounds ranging from approximately 40 Hz to 60 kHz. This ability allows them to respond to a variety of sounds, including bells. Most dogs are often trained from a young age to respond to different sounds which might signify food time, danger, or other cues in their environment. Their response can be very immediate, especially if the sound of the bell has been used as a training cue.
2. Understanding Grey Seal Hearing and Response: Grey seals have good hearing but are more adapted to detecting sounds under water rather than in the air. Their hearing range is roughly from 1 kHz to 70 kHz underwater, and they are less sensitive to sounds when they are on land. While Grey seals can be trained to respond to auditory stimuli in rehabilitation or research settings, this is less common compared to domestic dogs. Their general response time to sounds not associated with their immediate aquatic environments may be slower compared to dogs.
3. Comparing Response Times: Given that dogs are commonly trained to respond to auditory cues like bells and tend to have a quick response as part of their domestic training, they are likely to react faster. Grey seals, while capable of learning, are not typically exposed to the same consistent training regimens involving bells, especially in their natural environments.
4. Conclusion: Considering the usual domestic training of dogs related to sound cues and the comparative lack of such direct training with Grey seals, along with the differences in auditory sensitivities on land, a dog is more likely to respond to a bell before a Grey seal.
The answer is: Yes, a dog would respond to a bell before a Grey seal.

Table 55: Example 1: Using RAG to solve the first hard-proxy query in LBS3-(GSM8K, $n_1 = 1$)

---

**Input:**
Given examples:

Q: John read 20 pages of a book in the morning, and 30 pages in the afternoon. How many pages did John read in total?
A: Let's think step by step.
First, we know John read 20 pages in the morning.
Next, he read an additional 30 pages in the afternoon.
To find the total number of pages John read, we need to add the pages from the morning and afternoon together:
20 pages + 30 pages = 50 pages

The answer is 50 pages.

Solve the following problem based on given example above and present the final answer after "The answer is".

Q: Marcus baked cookies for 36 of his classmates in March, and then he baked twice as many cookies in April. How many cookies did Marcus bake altogether in March and April?
A: Let's think step by step.

---

**Output:**
First, we know Marcus baked 36 cookies for his classmates in March.
Next, we are told that Marcus baked twice as many cookies in April. To find how many cookies he baked in April, we need to multiply the number of cookies he baked in March by 2:
36 cookies × 2 = 72 cookies.

To find the total number of cookies Marcus baked in both March and April, we need to add the cookies he baked in March and April together:
36 cookies + 72 cookies = 108 cookies.

The answer is 108 cookies.

---

Table 56: Example 1: Using RAG to solve the second hard-proxy query in LBS3-(GSM8K, $n_1 = 1$)

---

**Input:**
Given examples:

Q: John read 20 pages of a book in the morning, and 30 pages in the afternoon. How many pages did John read in total?
A: Let's think step by step.
First, we know John read 20 pages in the morning.
Next, he read an additional 30 pages in the afternoon.
To find the total number of pages John read, we need to add the pages from the morning and afternoon together:
20 pages + 30 pages = 50 pages

The answer is 50 pages.

Q: Marcus baked cookies for 36 of his classmates in March, and then he baked twice as many cookies in April. How many cookies did Marcus bake altogether in March and April?
A: Let's think step by step. First, we know Marcus baked 36 cookies for his classmates in March. Next, we are told that Marcus baked twice as many cookies in April. To find how many cookies he baked in April, we need to multiply the number of cookies he baked in March by 2:
36 cookies × 2 = 72 cookies.

To find the total number of cookies Marcus baked in both March and April, we need to add the cookies he baked in March and April together:
36 cookies + 72 cookies = 108 cookies.

The answer is 108 cookies.

Solve the following problem step by step based on given examples above and present the final answer after "The answer is".

Q: Emily read books to 24 children in January, and then she read to three times as many children in February. How many children did Emily read books to altogether in January and February?
A: Let's think step by step.

---

**Output:**
First, we know Emily read books to 24 children in January.
Next, we are told that Emily read to three times as many children in February. To find out how many children that was, we need to multiply the number of children she read to in January by 3:
24 children × 3 = 72 children.

To find the total number of children Emily read books to in both January and February, we need to add the numbers for January and February together: 24 children + 72 children = 96 children.

The answer is 96 children.

---

Table 57: Example 1: Using RAG to solve the target query in LBS3-(GSM8K, $n_1 = 1$)

**Input:**
Given examples:

Q: John read 20 pages of a book in the morning, and 30 pages in the afternoon. How many pages did John read in total?
A: Let's think step by step.
First, we know John read 20 pages in the morning.
Next, he read an additional 30 pages in the afternoon.
To find the total number of pages John read, we need to add the pages from the morning and afternoon together:
20 pages + 30 pages = 50 pages

The answer is 50 pages.

Q: Marcus baked cookies for 36 of his classmates in March, and then he baked twice as many cookies in April. How many cookies did Marcus bake altogether in March and April?
A: Let's think step by step. First, we know Marcus baked 36 cookies for his classmates in March. Next, we are told that Marcus baked twice as many cookies in April. To find how many cookies he baked in April, we need to multiply the number of cookies he baked in March by 2:
36 cookies × 2 = 72 cookies.

To find the total number of cookies Marcus baked in both March and April, we need to add the cookies he baked in March and April together:
36 cookies + 72 cookies = 108 cookies.

The answer is 108 cookies.

Q: Emily read books to 24 children in January, and then she read to three times as many children in February. How many children did Emily read books to altogether in January and February?
A: Let's think step by step. First, we know Emily read books to 24 children in January.
Next, we are told that Emily read to three times as many children in February. To find out how many children that was, we need to multiply the number of children she read to in January by 3:
24 children × 3 = 72 children.

To find the total number of children Emily read books to in both January and February, we need to add the numbers for January and February together: 24 children + 72 children = 96 children.

The answer is 96 children.

Solve the following problem based on given examples above and present the final answer after "The answer is".

Q: Natalia sold clips to 48 of her friends in April, and then she sold half as many clips in May. How many clips did Natalia sell altogether in April and May?
A: Let's think step by step.

**Output:**
First, we know Natalia sold 48 clips to her friends in April.
Next, we are told that Natalia sold half as many clips in May. To find out how many clips that was, we need to divide the number of clips she sold in April by 2:
48 clips ÷ 2 = 24 clips.

To find the total number of clips Natalia sold in both April and May, we need to add the clips she sold in April and May together:
48 clips + 24 clips = 72 clips.

The answer is 72 clips.

Table 58: Example 2: Generation of easy- and hard-proxy queries by SPG and APG in LBS3 - (SQA, $n_1 = 2$)

---

**Input:**
Generate two different new relevant problems that are easier to solve than the example problem below. And then generate one new problem that is analogous to the example problem below.

Example problem:
Q: Would a dog respond to bell before Grey seal?

New problem 1:
Q:

---

**Output:**
Q: Does a cat meow more often than a dog barks?

New problem 2:
Q: Can a rabbit run faster than a turtle?

Analogous problem:
Q: Would a cat react to a whistle before a dolphin?

---

Table 59: Example 2: Using RAG to solve the first easy-proxy query in LBS3-(SQA, $n_1 = 2$)

---

**Input:**
Solve the following example problem and present the final answer after "The answer is".

Example problem:
Q: Does a cat meow more often than a dog barks?
A: Let's think step by step.

---

**Output:**
To solve this question, we need to consider the typical behaviors of cats and dogs, particularly focusing on their vocalization habits.
1. Understanding Cat Behavior: Cats meow for various reasons, such as to greet people, to seek attention, to ask for food, or to indicate distress. However, adult cats typically do not meow at each other, but rather reserve this sound for communicating with humans. The frequency of meowing can vary widely depending on the individual cat's personality, age, and circumstances.
2. Understanding Dog Behavior: Dogs bark as a form of communication with both humans and other dogs. The reasons for barking can include alerting to danger, expressing excitement, seeking attention, responding to other dogs, or as a response to boredom or anxiety. Like cats, the frequency of barking varies widely among individual dogs, influenced by breed, training, environment, and temperament.
3. Comparing the Two: Generally, dogs are considered to be more vocal than cats. Dogs use barking as a primary means of communication, and it is a behavior reinforced by both natural instincts and human training. Cats, while they do meow to communicate with humans, often use subtler forms of communication (like body language and scent marking) when interacting with other cats or when expressing themselves in their environment.
4. Conclusion: While there are exceptions depending on individual behavior and circumstances, generally speaking, dogs bark more frequently than cats meow. Dogs are often encouraged to bark for training and protective purposes, whereas cats typically meow less and often only in specific contexts (like interacting with humans).
The answer is: No, generally a cat does not meow more often than a dog barks.

---

Table 60: Example 2: Using RAG to solve the second easy-proxy query in LBS3-(SQA, $n_1 = 2$)

**Input:**
Solve the following example problem and present the final answer after "The answer is".

Example problem:
Q: Can a rabbit run faster than a turtle?
A: Let's think step by step.

**Output:**
To determine whether a rabbit can run faster than a turtle, we should consider the general known speeds of both animals.
1. Speed of a Rabbit: Rabbits are known for their ability to run quite fast to escape predators. The average speed of a wild rabbit is about 25-30 miles per hour (mph). Some rabbits can reach these top speeds so that they can quickly escape from threats.
2. Speed of a Turtle: Turtles, on the other hand, are known for their slow movement. The average walking speed of a typical turtle is approximately 0.2-0.5 miles per hour. Even when they feel threatened, turtles do not speed up significantly but rather retreat into their shells for protection.
3. Comparison: Comparing the speeds, 25-30 mph for rabbits vastly exceeds the 0.2-0.5 mph speed of a turtle.
Therefore, by analyzing the average speeds of both animals, it is clear that a rabbit can indeed run much faster than a turtle.
The answer is: Yes, a rabbit can run faster than a turtle.

Table 63: Example 3: Generation of easy- and hard-proxy queries by SPG and APG in LBS3 - (BBH-raco, $n_1 = 2$)

**Input:**
Generate two different new relevant problems that are easier to solve than the example problem below. And then generate one new problem that is analogous to the example problem below.

Example problem:
Q: On the desk, you see a set of things arranged in a row: a grey cup, a purple mug, and a blue teddy bear. What is the color of the thing directly to the right of the cup?Options: (A)red (B)orange (C)yellow (D)green (E)blue (F)brown (G)magenta (H)fuchsia (I)mauve (J)teal (K)turquoise (L)burgundy (M)silver (N)gold (O)black (P)grey (Q)purple (R)pink

New problem 1:
Q:

**Output:**
In a lineup of stationery items on your desk, you see a red stapler, a yellow pencil, and a green marker. What is the color of the item placed between the stapler and the marker? Options: (A)blue (B)yellow (C)orange (D)black

New problem 2:
Q: On the kitchen counter, there are three fruit bowls: one with oranges, one with apples, one with bananas. Which bowl is to the right of the bowl with apples? Options: (A)oranges (B)bananas (C)grapes (D)strawberries

Analogous problem:
Q: On the shelf, you see a series of items lined up: a red book, a green vase, and a yellow lamp. What is the color of the item directly to the left of the lamp? Options: (A) red (B) orange (C) yellow (D) green (E) blue (F) brown (G) magenta (H) fuchsia (I) mauve (J) teal (K) turquoise (L) burgundy (M) silver (N) gold (O) black (P) grey (Q) purple (R) pink

Table 64: Example 3: Using RAG to solve the first easy-proxy query in LBS3-(BBH-raco, $n_1 = 2$)

**Input:**
Solve the following example problem and present the final answer after "The answer is".

Example problem:
Q: In a lineup of stationery items on your desk, you see a red stapler, a yellow pencil, and a green marker. What is the color of the item placed between the stapler and the marker? Options: (A)blue (B)yellow (C)orange (D)black
A: Let's think step by step.

**Output:**
1. Identify the items mentioned in the lineup: a red stapler, a yellow pencil, and a green marker.
2. Determine the order of the items based on the question. The question asks for the color of the item placed between the stapler and the marker.
3. Since the only three items mentioned are the red stapler, the yellow pencil, and the green marker, and we need to find the item between the other two, we can deduce the order.
4. If the red stapler and the green marker are at the ends, the yellow pencil must be the item in between.
The answer is (B) yellow.

Table 65: Example 3: Using RAG to solve the second easy-proxy query in LBS3-(BBH-raco, $n_1 = 2$)

**Input:**
Solve the following example problem and present the final answer after "The answer is".

Example problem:
Q: On the kitchen counter, there are three fruit bowls: one with oranges, one with apples, one with bananas. Which bowl is to the right of the bowl with apples? Options: (A)oranges (B)bananas (C)grapes (D)strawberries
A: Let's think step by step.

**Output:**
To solve this problem, we need to determine the relative positions of the fruit bowls on the kitchen counter. The question specifically asks which bowl is to the right of the bowl with apples. Given the information in the question, we know there are three bowls containing: 1. Oranges 2. Apples 3. Bananas
The options provided are: (A) oranges (B) bananas (C) grapes (D) strawberries
Since the options for grapes and strawberries are not mentioned as being in any of the bowls on the counter, we can eliminate options (C) and (D).
Now, we need to decide between oranges and bananas. Since the question does not specify the exact order of the bowls, we have to make an assumption or use logical reasoning based on the given options. We know that one of the bowls to the right of the apples must be either the oranges or the bananas.
Without additional information about the arrangement, we can't definitively say which bowl is to the right of the apples. However, since we need to choose from the available options and assuming a typical left-to-right arrangement (if you face the counter), let's assume: - If the apples are in the middle, then the bananas could be to the right of the apples.
Thus, based on this assumption: The answer is (B) bananas.

Table 66: Example 3: Using RAG to solve the first hard-proxy query in LBS3-(BBH-raco, $n_1 = 2$)

**Input:**
Given examples:
Q: In a lineup of stationery items on your desk, you see a red stapler, a yellow pencil, and a green marker. What is the color of the item placed between the stapler and the marker? Options: (A)blue (B)yellow (C)orange (D)black
A: Let's think step by step.
1. Identify the items mentioned in the lineup: a red stapler, a yellow pencil, and a green marker.
2. Determine the order of the items based on the question. The question asks for the color of the item placed between the stapler and the marker.
3. Since the only three items mentioned are the red stapler, the yellow pencil, and the green marker, and we need to find the item between the other two, we can deduce the order.
4. If the red stapler and the green marker are at the ends, the yellow pencil must be the item in between.
The answer is (B) yellow.

Q: On the kitchen counter, there are three fruit bowls: one with oranges, one with apples, one with bananas. Which bowl is to the right of the bowl with apples? Options: (A)oranges (B)bananas (C)grapes (D)strawberries
A: Let's think step by step. To solve this problem, we need to determine the relative positions of the fruit bowls on the kitchen counter. The question specifically asks which bowl is to the right of the bowl with apples.
Given the information in the question, we know there are three bowls containing: 1. Oranges 2. Apples 3. Bananas
The options provided are: (A) oranges (B) bananas (C) grapes (D) strawberries
Since the options for grapes and strawberries are not mentioned as being in any of the bowls on the counter, we can eliminate options (C) and (D).
Now, we need to decide between oranges and bananas. Since the question does not specify the exact order of the bowls, we have to make an assumption or use logical reasoning based on the given options. We know that one of the bowls to the right of the apples must be either the oranges or the bananas.
Without additional information about the arrangement, we can't definitively say which bowl is to the right of the apples. However, since we need to choose from the available options and assuming a typical left-to-right arrangement (if you face the counter), let's assume: - If the apples are in the middle, then the bananas could be to the right of the apples.
Thus, based on this assumption: The answer is (B) bananas.

Solve the following problem based on given examples above and present the final answer after "The answer is".

Q: On the shelf, you see a series of items lined up: a red book, a green vase, and a yellow lamp. What is the color of the item directly to the left of the lamp? Options: (A)red (B)orange (C)yellow (D)green (E)blue (F)brown (G)magenta (H)fuchsia (I)mauve (J)teal (K)turquoise (L)burgundy (M)silver (N)gold (O)black (P)grey (Q)purple (R)pink
A: Let's think step by step.

**Output:**

1. Identify the items mentioned in the lineup: The question mentions three items: a red book, a green vase, and a yellow lamp.

2. Determine the order of the items: The question asks for the color of the item directly to the left of the lamp.

- We are given the following items in the question: - A red book - A green vase - A yellow lamp

3. Figure out the positioning: Since there are only three items mentioned, we need to figure out the relative positions. The question specifies that we are looking for the item to the left of the lamp, which implies that the lamp cannot be at the far left of the lineup. Therefore, the lamp must be the last item (to the right) in the sequence.

4. Establish the order: If the yellow lamp is the last item on the shelf, the other two items must be positioned to the left of it. The remaining items are the red book and the green vase.

5. Identify the item directly to the left of the lamp: Since the red book and the green vase are the only two items left, and we are looking for the one "directly to the left" of the yellow lamp, the green vase must be the item positioned to the left of the lamp.

The answer is (D) green.

Table 67: Example 3: Using RAG to solve the target query in LBS3-(BBH-raco, $n_1 = 2$)

**Input:**
Given examples:
Q: In a lineup of stationery items on your desk, you see a red stapler, a yellow pencil, and a green marker. What is the color of the item placed between the stapler and the marker? Options: (A)blue (B)yellow (C)orange (D)black
A: Let's think step by step.
1. Identify the items mentioned in the lineup: a red stapler, a yellow pencil, and a green marker.
2. Determine the order of the items based on the question. The question asks for the color of the item placed between the stapler and the marker.
3. Since the only three items mentioned are the red stapler, the yellow pencil, and the green marker, and we need to find the item between the other two, we can deduce the order.
4. If the red stapler and the green marker are at the ends, the yellow pencil must be the item in between.
The answer is (B) yellow.

Q: On the kitchen counter, there are three fruit bowls: one with oranges, one with apples, one with bananas. Which bowl is to the right of the bowl with apples? Options: (A)oranges (B)bananas (C)grapes (D)strawberries
A: Let's think step by step. To solve this problem, we need to determine the relative positions of the fruit bowls on the kitchen counter. The question specifically asks which bowl is to the right of the bowl with apples.
Given the information in the question, we know there are three bowls containing: 1. Oranges 2. Apples 3. Bananas
The options provided are: (A) oranges (B) bananas (C) grapes (D) strawberries
Since the options for grapes and strawberries are not mentioned as being in any of the bowls on the counter, we can eliminate options (C) and (D).
Now, we need to decide between oranges and bananas. Since the question does not specify the exact order of the bowls, we have to make an assumption or use logical reasoning based on the given options. We know that one of the bowls to the right of the apples must be either the oranges or the bananas.
Without additional information about the arrangement, we can't definitively say which bowl is to the right of the apples. However, since we need to choose from the available options and assuming a typical left-to-right arrangement (if you face the counter), let's assume: - If the apples are in the middle, then the bananas could be to the right of the apples.
Thus, based on this assumption: The answer is (B) bananas.

Q: On the shelf, you see a series of items lined up: a red book, a green vase, and a yellow lamp. What is the color of the item directly to the left of the lamp? Options: (A)red (B)orange (C)yellow (D)green (E)blue (F)brown (G)magenta (H)fuchsia (I)mauve (J)teal (K)turquoise (L)burgundy (M)silver (N)gold (O)black (P)grey (Q)purple (R)pink
A: Let's think step by step. 1. Identify the items mentioned in the lineup: The question mentions three items: a red book, a green vase, and a yellow lamp.
2. Determine the order of the items: The question asks for the color of the item directly to the left of the lamp.
- We are given the following items in the question: - A red book - A green vase - A yellow lamp
3. Figure out the positioning: Since there are only three items mentioned, we need to figure out the relative positions. The question specifies that we are looking for the item to the left of the lamp, which implies that the lamp cannot be at the far left of the lineup. Therefore, the lamp must be the last item (to the right) in the sequence.
4. Establish the order: If the yellow lamp is the last item on the shelf, the other two items must be positioned to the left of it. The remaining items are the red book and the green vase.
5. Identify the item directly to the left of the lamp: Since the red book and the green vase are the only two items left, and we are looking for the one "directly to the left" of the yellow lamp, the green vase must be the item positioned to the left of the lamp.
The answer is (D) green.

Solve the following problem based on given examples above and present the final answer after "The answer is".
Q: On the desk, you see a set of things arranged in a row: a grey cup, a purple mug, and a blue teddy bear. What is the color of the thing directly to the right of the cup?Options: (A)red (B)orange (C)yellow (D)green (E)blue (F)brown (G)magenta (H)fuchsia (I)mauve (J)teal (K)turquoise (L)burgundy (M)silver (N)gold (O)black (P)grey (Q)purple (R)pink
A: Let's think step by step.

**Output:**
1. Identify the items mentioned in the lineup: The question mentions three items: a grey cup, a purple mug, and a blue teddy bear.
2. Determine the order of the items: The question asks for the color of the thing directly to the right of the cup.
3. Figure out the positioning: Since the question specifies that we are looking for the item directly to the right of the grey cup, the cup cannot be at the far right of the lineup. Therefore, the cup must be positioned either at the far left or in the middle.
4. Establish the order: Given that there are only three items and the cup needs to have an item to its right, the possible positions for the cup are either first or second in the lineup. The remaining items are the purple mug and the blue teddy bear.
5. Identify the item directly to the right of the cup: Since the options provided include colors that match the items mentioned (purple and blue), and we need to find the item directly to the right of the grey cup, we can deduce that the purple mug or the blue teddy bear could be next to it.
6. Choose the correct option based on available colors: Since the colors purple and blue are both options and we need to select one, we assume a typical left-to-right arrangement (if you face the desk). If the grey cup is first, then the next item (to the right) could logically be the purple mug, as it fits the sequence and matches one of the available color options.
The answer is (Q) purple.

