# OpenReview forum: "Let's Be Self-generated via Step by Step: A Curriculum Learning Approach to Automated Reasoning with Large Language Models"
_ICLR.cc/2025/Conference — ICLR 2025 Conference Withdrawn Submission_

### Official Review · Reviewer_8bhm · 2024-11-02

**Soundness:** 3
**Presentation:** 3
**Contribution:** 2
**Rating:** 5
**Confidence:** 4

**Summary:**

This paper presents a new LLM prompting technique: given a problem (e.g., math), generate easy examples (phase 1), generate hard examples (phase 2), and finally solve the problem conditioned on examples. The key insight is that previous approaches to automatically generating examples might generate examples that are too hard, which isn't as helpful, so one can think of this work as bringing curriculum learning ideas into the prompting regime.

**Strengths:**

- The idea of easy-to-hard prompting is natural, the solution is simple and elegant, which is well appreciated.
 - For the experiments, the choice of models, datasets, and baselines are all reasonably thorough and solid.

**Weaknesses:**

- I feel like the paper could have been a bit stronger on the analysis. The paper does ablations on the number of examples used in each of the two stages (Figure 3), examines using only easy or only hard examples (Figure 5), and compares against the baselines. It demonstrates the effectiveness of the method but it doesn't really tell me why the method is effective. For example, I would be interested in the type of errors that are made as result of the suboptimal ablations; can the reason for failure be attributed to bad reasoning? How valid are the examples generated? Are the easy examples actually easy and the harder examples actually harder? What if more exemplars were used?  How sensitive is is the method to the wording of the prompt (how was the prompt in Table 3 derived?)?
 - Moreover, the fact that Llama3-70B-Instruct is better than GPT-4.0 turbo on most tasks is very counterintuitive. Might it be some artifact of the prompting?  It just seems that GPT-4.0 turbo has to be a stronger model (according to every benchmark I've seen). This definitely warrants some explanation, but there wasn't anything satisfying provided.

**Questions:**

- Line 310: for closed-source models, why can't you set the temperature = 0 to get determinism?
 - Section 4.2: why does Llama3-70B-Instruct outperform GPT-4.0 turbo?  "We considered only a limited set of reasoning tasks" is given as an explanation, but I find this rather unsatisfactory, because GPT-4.0 turbo should really dominate Llama3-70B across the board. This makes me think that something is wrong.

---

> ### Author Response · Authors · 2024-11-23
> **Response to Reviewer 8bhm (1/2)**
>
> Thanks for your valuable review and suggestions! It is encouraging to see you find our methodology simple, elegant and well appreciated.
> We sincerely thank you for your time and constructive comments. Below, we provide detailed replies to your comments to resolve your concerns.
>
> **Q1:** I feel like the paper could have been a bit stronger on the analysis. The paper does ablations on the number of examples used in each of the two stages (Figure 3), examines using only easy or only hard examples (Figure 5), and compares against the baselines. It demonstrates the effectiveness of the method but it doesn't really tell me why the method is effective. For example, I would be interested in the type of errors that are made as result of the suboptimal ablations; can the reason for failure be attributed to bad reasoning? How valid are the examples generated? Are the easy examples actually easy and the harder examples actually harder? What if more exemplars were used? How sensitive is is the method to the wording of the prompt (how was the prompt in Table 3 derived?)?
>
> **R1:** Thank you very much for your comment. Below, we will respond to each point individually. Specifically,
>
> - **1.** "I would be interested in the type of errors that are made as result of the suboptimal ablations; can the reason for failure be attributed to bad reasoning?"
>
> We argue that the failure of suboptimal ablations may be attributed to poor reasoning, but more to the number of easy- and hard-proxy examplars compared to optimal ablation.
> For example, in Figure 3 (b), we consider the experimental results of the benchmark GSM8K and the model Llama3-70-Instruct. When $n_1>0$ and $n_2>0$, the optimal ablation result was 94.6\% ($n_1=2$) and the suboptimal ablation result was 93.4\%($n_1=1$). It can be seen that the above results are obtained from experiments on the same model and benchmark, and other settings are the same except $n_1$ and $n_2$ (note that $n=n_1+n_2$). This ensures that for the same query, the difference in results comes from the ratio of the number of easy- and hard-proxy examplars. Therefore, we believe that solving more error prone hard-proxy queries (i.e. $n_1=1$) results in incorrect target query solutions.  For ease of explanation, we have further analyzed the quality of the proxy queries generated in LBS3 and provided several complete reasoning processes in the revised manuscript.  **Please refer to section 4.3.2 of the revised paper for details.**
>
> - **2.** "How valid are the examples generated?"
>
> Regarding the effectiveness of the generated examples, we conducted a comparative analysis in Section 4.2 and fully verified it, which is also illustrated by the real example provided in Figure 1. In addition, we provide specific experiments in section 4.3.2 of the revised paper to support this point.
>
> - **3.** "Are the easy examples actually easy and the harder examples actually harder?"
>
> You can refer to the large number of examples provided in Appendix C of the manuscript to answer this question with certainty. The prompt framework we propose can achieve clear distinctions between easy- and hard-proxy queries on different LLMs. To further confirm this conclusion, we conducted supplementary research in the revised version of the paper.  **Please refer to section 4.3.2 of the revised paper for details.**
>
> - **4.** "What if more exemplars were used? "
>
> This is an open question. In our experiments, we fixed the number of prompts to 3 or 4. On one hand, we followed the experimental setups of existing methods such as Self-ICL, Auto-ICL, and Ana-Pro. On the other hand, within the constraints of effective runtime and cost, our setup has yielded satisfactory experimental outcomes. Therefore, we have forgone the exploration for the optimal results, reserving this for future work.
>
> - **5.** "How sensitive is the method to the wording of the prompt (how was the prompt in Table 3 derived?)?"
>
> The proposed method has different sensitivities to the wording of prompts for different sizes of LLMs. Specifically, in the specific experimental process, we made carefully adjustments to the key wording and expressions in SPG and APG in terms of simplicity, effectiveness, and universality (mainly including the LLMs used in this study). For example, when replacing synonyms for keywords (e.g. relevant，easier, solve and analogous), and trying other expressions (see Tables 4-6), it was found that on smaller models (such as Qwen1.5-14B Chat), easy- and hard-proxy queries could not be generated correctly. Similarly, this can be confirmed in Appendix C. Therefore, we can ensure that the prompts reported in LBS3 are effective on the five LLMs in this work.

---

> ### Author Response · Authors · 2024-11-23
> **Response to Reviewer 8bhm (2/2)**
>
> **Q2:** Moreover, the fact that Llama3-70B-Instruct is better than GPT-4.0 turbo on most tasks is very counterintuitive. Might it be some artifact of the prompting? It just seems that GPT-4.0 turbo has to be a stronger model (according to every benchmark I've seen). This definitely warrants some explanation, but there wasn't anything satisfying provided.
>
> **R2:**  We agree with your point. We also expressed the same doubts in the manuscript and provided our guesses and explanations, please refer to lines 332-337. Simply put, we argue that GPT-4.0 turbo performs better overall than Llama3-70B-Instruction on more tasks. However, since the limited inference tasks considered in this study, there is a counterintuitive situation where the overall performance of Llama3-70B-Instruction is slightly better than that of GPT-4.0 turbo. Due to considerations of calculation, time and cost, as well as alignment with existing works (e.g., Self-ICL, Auto-ICL and Ana-Pro), we focus on the comparison between LBS3 and baseline methods. In other word, The goal of our research was to propose an effective automatic reasoning prompt method for Large Language Model (LLM) to more effectively tap into the reasoning capabilities of LLMs. Additionally, we can assure you that the results reported are accurate within the specific context of our experiments. Notably, we believe this is important and not-trivial, so we will delve deeper into it in our future work. We hope this clarification addresses your concern.
>
> **Q3:** Line 310: for closed-source models, why can't you set the temperature = 0 to get determinism?
>
> **R3:** Thank you for your comment. For open-source models, we use greedy algorithms to ensure the certainty of the experimental process. This is because when the temperature is set to 0, although it can make the LLM output more stable, there is still randomness.  For the closed-source models, we set the temperature to 0 for the experiments and report the average results of three runs.
>
> **Q4:** Section 4.2: why does Llama3-70B-Instruct outperform GPT-4.0 turbo? "We considered only a limited set of reasoning tasks" is given as an explanation, but I find this rather unsatisfactory, because GPT-4.0 turbo should really dominate Llama3-70B across the board. This makes me think that something is wrong.
>
> **R4:** Please refer to **R2** for details.
>
> We hope this clarification can address your concerns and look forward to further discussions with you.

---

> ### Author Response · Authors · 2024-11-25
>
> Dear Reviewer 8bhm,
> ﻿
> We would like to express my sincere gratitude for the time and effort you have dedicated to evaluating our submission.
> ﻿
> As the rebuttal is nearing its conclusion, if you have any further questions or require additional clarification, please do not hesitate to reach out to us. We sincerely hope that our responses satisfactorily resolve your queries and that you might consider revising your score in light of the clarifications provided.
> ﻿
> Thank you once again for your thoughtful review.
> ﻿
> Sincerely,
> ﻿
> Authors of Paper 5493

---

### Official Review · Reviewer_FEX4 · 2024-11-04

**Soundness:** 3
**Presentation:** 3
**Contribution:** 3
**Rating:** 6
**Confidence:** 4

**Summary:**

This paper proposed a way for LLM prompting. It first uses a two-stage generation of proxy queries and a way to progressively solve the proxy queries. The method is shown to outperform many baselines in different complex question reasonly tasks.

**Strengths:**

* The prompting method outperforms many baselines for solving complex tasks for LLM, verified under different LLMs
* The method makes sense intuitively.

**Weaknesses:**

* More ablation studies are needed to show the components proposed in this paper are necessary

**Questions:**

The reviewer believes the biggest question related to this paper is the lack of ablation. The reviewers should show the model performances for
* What happens if we don't include APG questions? (aka, n2=0)
* What happens if we don't generate questions using LLM, but use some gold examples in the dataset?
* Does the order of examples in the final RAG-F stages matter, if we separates easy examples and hard examples?
* For the final RAG-F, can we only retrieve hard examples?

---

> ### Author Response · Authors · 2024-11-23
> **Response to Reviewer FEX4 (1/2)**
>
> Thanks for your valuable review and suggestions! It is encouraging to see you find our methodology well-preformanced and reasonable .
> We sincerely thank you for your time and constructive comments. Below, we provide detailed replies to your comments to resolve your concerns.
>
> **Q1:** What happens if we don't include APG questions? (aka, $n_2=0$)
>
> **R1:**  We have taken into account your concern in the experimental section of the manuscript. Specifically, in the experimental section 4.3.1, we conducted thorough experiments on multiple models and benchmarks to verify the effects of different $n_1$ and $n_2$ settings on LBS3. According to our experimental setup (see lines 306 to 309), $n_2=n-n_1$. Therefore, when $n_1=3$, the corresponding $n_2=0 $. For the sake of conciseness, we only provide the variation of the value of $n_1$, as shown in Figure 3.
>
> **Q2:**  What happens if we don't generate questions using LLM, but use some gold examples in the dataset?
>
> **R2:**   We considered this point in the comparative study in Section 4.2. Specifically, the baseline method we provide, few shot CoT, uses some golden examples from the dataset for reasoning.  And we provide experimental results as shown in Table 1 of the paper.
> However, we argue that the Reviewer might be more interested in understanding the performance changes that could arise from using some golden examples within the framework of LBS3. To this end, we conducted some preliminary experiments on MATH and SVMP using Qwen1.5-14B-Chat. During the experimental phase, we selected some golden examples from the dataset to directly replace the query inputs in RAG-Z and RAG-F, while keeping them unchanged to avoid LLM from generating proxy queries. To maintain a fair comparison, we select four examples and assigned two examples each to RAG-Z and RAG-F in order of increasing difficulty. We call this baseline as **LBS3-Few-shot** and report the experimental results in the table below.
>
> | Method | **MATH** | **SVMP** |
> | --- | --- | --- |
> | LBS3-Few-shot | 37.95 | 84.62 |
> | Few-shot | 36.8 | 84.4 |
> | LBS3 | **40.8** | **85.8** |
>
> From above Table, we find that LBS3-Few-shot **underperforms** LBS3 in terms of performance, but it dose achieve better results than Few-shot. We argue that the effectiveness of LBS3 relies on Progressive Strategy, which aims to avoid starting from scratch in solving hard-proxy queries, thereby ensuring the more accurate generation of solutions for hard-proxy queries. Moreover, the said results indicate that the capability of LBS3-Few-shot to assist in answering hard-proxy queries with fixed manual prompt exemplars may be less effective than LBS3's use of analogous easy-proxy exemplars and already generated hard-proxy exemplars.
>
> Although we have provided **extra experiments**, it can be foreseen that retrieving suitable golden samples from existing datasets for each target query will further improve performance. However, doing so would violate the original intention of our work, and it pertains to another independent and systematic research topic, as shown in lines 48 to 57. The goal of our research was to propose an effective automatic reasoning prompt method for Large Language Model (LLM) to more effectively tap into the reasoning capabilities of LLMs. We hope this clarification addresses your concern.

---

> ### Author Response · Authors · 2024-11-23
> **Response to Reviewer FEX4 (2/2)**
>
> **Q3:** Does the order of examples in the final RAG-F stages matter, if we separates easy examples and hard examples?
>
> **R3:** We deem this is important. However, as far as we know, the study of example order is an independent research topic, and there have been some research works on it **[1],[2],[3]**. In our experimental setup, LBS3 has shown superior performance compared to the baseline methods (see Table 1), so the study of example order is not our focus. We will consider the impact of changes in the order of RAG-F stage examples on the performance of LBS3 in future research.
>
> But to address the reviewers' concerns, we provide a preliminary exploratory experiment. To be specific, we conducted experiments again using Qwen1.5-14B-Chat on MATH and SVAMP. The experimental setup corresponds with lines 301 to 310 of the paper, where $n_1 = 2$ and $n_2=2$. We assigned LBS3 to sequentially number the proxy queries generated for each target query from easiest to most difficult (stored locally): 1, 2, 3, 4, with 1 and 2 representing the numbers for simple proxy queries and 3 and 4 for complex ones. We report the experimental results in the table below.
>
> | Method | **MATH** | **SVMP** |
> | --- | --- | --- |
> | basline1 | 40.52 | 85.43 |
> | basline2 | 39.87 | 84.64 |
> | basline3 | 38.22 | 83.41 |
> | basline4 | 40.28 | 84.76 |
> | basline5 | 40.11 | 84.65 |
> | LBS3 | **40.80** | **85.80** |
>
> For the above Table, **baseline1** indicates a reordering to 2, 1, 4, 3; **baseline2** indicates a reordering to 3, 4, 1, 2; **baseline3** indicates a reordering to 4, 3, 2, 1; **baseline4** indicates a reordering to 1, 3, 2, 4; and **baseline5** indicates a reordering to 3, 1, 4, 2.
>
> From the Table, it is evident that after the shuffling of the proxy queries, the performance of LBS3 experienced varying degrees of decline. Notably, the significant drop in baselines 2 and 3 suggests that the overall interchange of simple and complex proxy queries may exert a considerable influence on the performance of Qwen1.5-14B-Chat.  The preliminary experimental results indicate that changes in the order of examples can affect reasoning performance. We hope this clarification addresses your concern.
>
> **[1]** Lu Y, Bartolo M, Moore A, et al. Fantastically ordered prompts and where to find them: Overcoming few-shot prompt order sensitivity[J]. arXiv preprint arXiv:2104.08786, 2021.
>
> **[2]** Guo Q, Wang L, Wang Y, et al. What makes a good order of examples in in-context learning[C]//Findings of the Association for Computational Linguistics ACL 2024. 2024: 14892-14904.
>
> **[3]** Xiang Y, Yan H, Gui L, et al. Addressing Order Sensitivity of In-Context Demonstration Examples in Causal Language Models[J]. arXiv preprint arXiv:2402.15637, 2024.
>
> **Q4:** For the final RAG-F, can we only retrieve hard examples?
>
> **R4:** Other examples can be retrieved. It's just that LBS 3 focuses on hard examples because this method setup can reduce the performance requirements for large models while maintaining sufficient simplicity.
>
> We hope this clarification can address your concerns and look forward to further discussions with you.

---

### Official Review · Reviewer_fHem · 2024-11-04

**Soundness:** 2
**Presentation:** 3
**Contribution:** 2
**Rating:** 6
**Confidence:** 3

**Summary:**

This paper introduces a novel automatic reasoning prompt approach called LBS3, inspired by the principles of curriculum learning. LBS3 aids large language models (LLMs) in recalling both easy and hard proxy queries related to a target query. Subsequently, it employs a progressive strategy that leverages exemplary prompts derived from easy proxy queries to guide LLMs in addressing hard proxy queries, thereby enhancing the quality of the proxy solutions. Experiments conducted across various reasoning-intensive tasks using both open-source and closed-source LLMs demonstrate that LBS3 achieves competitive performance.

**Strengths:**

1. The paper is well-written and easy to follow.

2. The performance of LBS3 is strong.

3. The idea of LBS3 is simple and effective.

**Weaknesses:**

1. Lack of theoretical contribution. Although the performance of LBS3 is quite promising, its technical contribution compared to existing methods (such as Ana-Pro) seems minor.

2. To enhance the paper's contribution, it would be advantageous to provide insights into how simpler exemplars can improve LLM's accuracy on more challenging exemplars.

**Questions:**

The analysis lacks a quantitative evaluation of the model's accuracy in responding to hard proxy exemplars, which would demonstrate whether the easy proxy exemplars generated by SPG actually help improve the LLM's performance on harder ones.

---

> ### Author Response · Authors · 2024-11-23
>
> Thanks for your valuable review and suggestions! It is encouraging to see you find our paper truly makes sense, and that the proposed method is simple, effective, and well-perfomanced.
> We sincerely thank you for your time and constructive comments. Below, we provide detailed replies to your comments to resolve your concerns.
>
> **Q1:** Lack of theoretical contribution. Although the performance of LBS3 is quite promising, its technical contribution compared to existing methods (such as Ana-Pro) seems minor.
>
> **R1:** We greatly appreciate your comment. Below, we will respond separately from theoretical and technical contributions.
>
> **For the former,**  we agree with your point that there is a lack of theoretical contribution. The black-box nature of Large Language Models (LLMs) makes it challenging for us to provide theoretical contributions. As far as we know, our research, similar to Self-ICL, Auto-ICL, and Ana-pro, is dedicated to the application side.  In an attempt to fill this gap, we have supplemented the revised paper with a detailed empirical study of the proxy queries generated by LBS3.  **Please refer to section 4.3.2 of the revised paper for details.**
>
> **For the latter,** we first described the existing method Ana Pro in the paper, as shown in lines 182-186. Simply put, the one-pass generation mode employed for Ana Pro necessitates that the LLM possesses robust capabilities for both following instructions and generating responses.
>
> We argue that LBS3 has the following advantages compared to Ana Pro:
>   -  LBS3 has put forward more moderate requirements for LLM's ability to follow instructions and generate responses through the two-stage (i.e., query generation and solution generation) manner.
>   -  LBS3 draws on curriculum learning to guide LLMs to autonomously generate proxy queries from easy to difficult and proposes Progressive Strategy to sequentially solve difficult proxy queries one by one, thereby addressing the main query. To our knowledge, no previous work has attempted to generate proxy queries from easy to difficult to assist in solving the target query. This is the main contribution of this work to the field of prompt engineering and also where its innovation lies.
>
> Therefore, we believe that LBS3 has significant technological differences, compared to Ana Pro.
>
> **Q2:** To enhance the paper's contribution, it would be advantageous to provide insights into how simpler exemplars can improve LLM's accuracy on more challenging exemplars.
>
> **R2:** We agree with your point. First, we have thoroughly discussed this point in the INTRODUCTION section of the manuscript in conjunction with existing work, as shown in lines 58-70, 143-150, and Figure 1. In addition, we provide specific experiments in section 4.3.2 of the revised paper to support this point.
>
> **Q3:** The analysis lacks a quantitative evaluation of the model's accuracy in responding to hard proxy exemplars, which would demonstrate whether the easy proxy exemplars generated by SPG actually help improve the LLM's performance on harder ones.
>
> **R3:** In section 4.3.1 of the manuscript's experimental section, we conducted thorough experiments on multiple models and benchmarks to verify the impact of different $n_1$ and $n_2$ settings on LBS3.  According to our experimental setup (see lines 306 to 309), $n_2 = n - n_1$. So, when $n_1=1$, the corresponding $n_2=2 $. For the sake of conciseness, we only provide the variation of the value of $n_1 $, as shown in Figure 3. Therefore, our study covers quantitative evaluations of the accuracy of the model in responding to hard proxy examples ($n_1=0,1,2$) and the role of SPG ($n_1=3$). The details can be found in lines 400 to 421 and Figure 3.
>
> We hope this clarification can address your concerns and look forward to further discussions with you.

---

> > ### Author Response · Authors · 2024-11-25
> >
> > Dear Reviewer fHem,
> > ﻿
> > We would like to express my sincere gratitude for the time and effort you have dedicated to evaluating our submission.
> > ﻿
> > As the rebuttal is nearing its conclusion, if you have any further questions or require additional clarification, please do not hesitate to reach out to us. We sincerely hope that our responses satisfactorily resolve your queries and that you might consider revising your score in light of the clarifications provided.
> > ﻿
> > Thank you once again for your thoughtful review.
> > ﻿
> > Sincerely,
> > ﻿
> > Authors of Paper 5493

---

> > ### Comment · Reviewer_fHem · 2024-11-27
> >
> > Thank you for your responses, and I apologize for the delayed reply. I have thoroughly reviewed the authors' responses and the revised paper. The additional empirical study in section 4.3.2 effectively addresses my concerns about whether the easy proxy exemplars can genuinely enhance the LLM's accuracy in generating solutions for the difficult proxy exemplars. I am convinced that LBS3 has the potential to improve the performance of LLMs on reasoning tasks and holds practical value. For the sake of that, I am increasing my score to 6.
> >
> > However, regarding the two advantages of LBS3 claimed by the authors, I still have some concerns.
> > - Firstly, the two-stage approach utilized by LBS3 is not particularly novel, as similar methods have been implemented in Self-ICL and Auto-ICL, which also generate a query first before producing the solution.
> > - Secondly, while I appreciate that the progressive strategy of LBS3 has not previously appeared in the literature, it strikes me as rather straightforward and easy to conceive.
> >
> > Given these points, I am uncertain whether the novelty of this paper meets the standards of ICLR and how much new insight it truly contributes to the field. As a result, I will decrease my confidence score to 3.
> >
> > Furthermore, the authors convey the message that the improved performance of LBS3 is due to the increased accuracy in generating solutions for the hard proxy exemplars (Please correct me if I misunderstood). However, there is currently no consensus on whether a CoT demonstration needs to be correct in order to be beneficial for the model when answering questions. As noted in [1], other aspects of the rationales, such as relevancy to the query and correct ordering of the reasoning steps, are much more important for effective CoT reasoning. Considering this, I am particularly interested in understanding the underlying reasons why the hard proxy solution, generated with the help of easy-proxy examples, can improve the model's ability to answer query questions. Nevertheless, since the authors have clearly indicated that this paper focuses on the application side, I will refrain from requesting any additional content on this topic.
> >
> > [1] Wang et al. Towards Understanding Chain-of-Thought Prompting: An Empirical Study of What Matters

---

> ### Author Response · Authors · 2024-11-28
>
> Dear Reviewer fHem,
>
> Thank you for taking the time to provide valuable feedback and for acknowledging our work. We would like to provide some additional explanations and details to clarify any misunderstandings between us.
>
> **Q1:** The two-stage approach utilized by LBS3 is not particularly novel, as similar methods have been implemented in Self-ICL and Auto-ICL, which also generate a query first before producing the solution.
>
> **R1:** Although Self-ICL and Auto-ICL are similar in form to LBS3, LBS3 has essential differences from them. Figure 1 significantly demonstrates the differences and superiority of LBS3. Simply put, the contribution of LBS 3 is mainly reflected in 1) effectively guiding LLM to generate easy- and hard-proxy queries related to the target query, and 2) improving the effectiveness of solving proxy queries, especially for hard-proxy queries, thereby enhancing the solution for the target queries. It is worth noting that we elaborated on the innovation of LBS3 and its differences from existing methods (Self-ICL and Auto-ICL) in detail from lines 97 to 117 and from lines 143 to 150.
>
> **Q2:** While I appreciate that the progressive strategy of LBS3 has not previously appeared in the literature, it strikes me as rather straightforward and easy to conceive.
>
> **R2:** Thank you for your appreciation of LBS3’s advancement strategy. Although it is simple, it does perform well on multiple reasoning tasks in zero-shot scenarios. This is also the significance of our work, which is to achieve excellent performance using simple strategies.
>
> **Q3:** The authors convey the message that the improved performance of LBS3 is due to the increased accuracy in generating solutions for the hard proxy exemplars (Please correct me if I misunderstood). However, there is currently no consensus on whether a CoT demonstration needs to be correct in order to be beneficial for the model when answering questions. As noted in [1], other aspects of the rationales, such as relevancy to the query and correct ordering of the reasoning steps, are much more important for effective CoT reasoning. Considering this, I am particularly interested in understanding the underlying reasons why the hard proxy solution, generated with the help of easy-proxy examples, can improve the model's ability to answer query questions.
>
> **R3:** Firstly, an intuitive point of LBS3 is to improve the solution of hard-proxy queries, thereby enhancing the accuracy of target queries. In most of our experimental scenarios, LBS3 showed performance improvement. For example, in Figure 3, based on the progressive strategy, in most cases, the performance of LBS3 with $n_1=0$ (i.e., generating only hard proxy queries) is inferior to that of LBS3 with $n_1=1$ or $n_1=2$. Meanwhile, according to the APG-acc metric in Table 2, the accuracy of hard-proxy queries was consistently improved on all datasets with the help of easy-proxy queries. From the above experience results, it can be concluded that easy-proxy queries improve the accuracy of hard-proxy query solutions, thereby enhancing the model's ability to answer target queries.
>
> In addition, we agree that the relevance of the query and the correct order of reasoning steps are crucial. They have received widespread attention as independent research topics. However, the contribution of LBS3 lies in the preliminary empirical exploration of whether correctness is necessary in answering questions. We believe that explaining and clarifying the underlying principles is important and highly challenging, and will leave it to future work.
>
> We hope our response has addressed your concerns, and we look forward to your further feedback.

---

### Author Response · Authors · 2024-11-23

Dear Reviewers,

We thank all reviewers for their thoughtful and constructive review of our manuscript, and acknowledgment of our contributions.

**We greatly appreciate your recognition of the strengths of our work as follows:**
- **Introduction of  LBS3**
  - We present the LBS3, as a novel approach, recognized by  **all reviewers as well-motivated.**
  -  Our approach offers a **simple, natural , reasonable and elegant** (fHem, FEX4, 8bhm) framework for solving complex reasoning tasks for LLMs, which is **well appreciated**（8bhm）.
- **Methodological Effectiveness**
  - Our approach has been acknowledged by the **reviewers** fHem and FEX4 for its **effectiveness and strong performance**, verified under different LLMs .
- **Comprehensive Experiments**
  - Our Experiments is acknowledged for its **comprehensiveness and solid** by the reviewer 8bhm, including the **choice of models, datasets, and baselines**. Meanwhile,  the reviewer fHem praises our paper for being well written.

**We've revised our manuscript for the reviewers' suggestions** (**highlighted in red** in the uploaded revision pdf). Detailed responses to each reviewer's concerns are carefully addressed point-by-point.

Below summarize the **major updates** we've made:
- **Experiment**: We further conduct the following experiments to make our paper more sound and try to address reviewers' concerns. Specifically, we investigated the quality of easy-  and hard-proxy queries generated in LBS3 and added **section 4.3.2** in the main paper, and accordingly added **Appendix D** (fHem, 8bhm).
- **Explanation**: Due to space constraints in the main paper, we moved the **Limitations** and **Broader Impacts** within **Conclusion** to Appendix **E** and **F**, respectively.

We believe our work could make a novel contribution to the community and offer a novel perspective on addressing the challenges of complex reasoning tasks of LLM.

We would like to be involved in further discussions if any question is raised.

Best,

Authors.

---

### Comment · Area_Chair_EkDi · 2024-11-24
**Reminder: Author-Reviewer Discussion Period Closing Soon**

This is a reminder that the author-reviewer discussion period will end on Nov 26 AoE.

Your engagement during this phase is critical for providing valuable feedback and clarifications. If you have any remaining questions or comments, please take a moment to participate before the deadline.

Thank you for your contributions to this important process.

AC

---

### Note · Authors · 2025-01-22

I have read and agree with the venue's withdrawal policy on behalf of myself and my co-authors.